# Pre-existing and early cellular immune factors correlate with functionally complete protection against primary controlled human SARS-CoV-2 infection

Helen R. Wagstaffe [1], Ryan S. Thwaites [2], Jasmin K. Sidhu[2], Rik G. H. Lindeboom [3,4], Lorenz Kretschmer[3,5], Kaylee B. Worlock [6], Lisa M. Dratva [3,5], Ao Huang[7,8], Stephanie Ascough[1], Loukas Papargyris [1], Richard McKendry[1], Ashley M. Collins [1], Jiayun Xu[1], Nana-Marie Lemm[1], Ben Killingley[9], Mariya Kalinova[10], Alex Mann [10], Andrew Catchpole [10], Leo Swadling [11], John S. Tsang [7,8,12], Mala K. Maini [11], Mahdad Noursadeghi[11], Marko Z. Nikolić [6], Sarah A. Teichmann [4,5], Peter J. M. Openshaw [2] & Christopher Chiu [1] ✉

Identifying host factors that mediate protection against newly-emergent viruses is needed for improved pandemic preparedness. Here, we analysed pre- and early post-exposure immune factors associated with resisting SARS-CoV-2 infection after human challenge in seronegative individuals, using multiplex protein, cytometric and RNA sequencing approaches in the naso-pharynx and circulation. Pre-existing cross-reactive antibodies correlate poorly with clinical outcome. Instead, protection is associated with heightened nasopharyngeal CCL13 levels locally produced by conventional dendritic cells and monocytes, along with cross-reactive T cells and less differentiated NK cells. Conditional independence network analysis implicates nasal CCL13 as the central node connected to pre-existing non-structural protein-specific T cells by CD1c[+] DCs. In those who became infected, baseline cross-reactive T cell and less differentiated NK cell frequencies also correlate with shorter infection duration. Thus, pre-existing mucosal chemokine levels may promote rapid innate and innate-like responses that effectively block infection. ClinicalTrials.gov identifier NCT04865237.

During the early months of the COVID-19 pandemic, anecdotal reports suggested that a proportion of individuals could have resisted primary infection following known exposure despite the absence of pre-existing strain-specific immunity. While most of the world's population have since been infected by SARS-CoV-2, especially as new variants have appeared, it remains likely that individual host-related variations in susceptibility against emergent viruses exist and that some people may be more resistant to infection than others at the time of first virus encounter.

In field studies, it is almost impossible to study resistors due to limitations in case detection and differences in viral, environmental and behavioural factors that cannot be adequately measured or controlled for. In contrast, the controlled nature of human challenge studies enables more robust investigation of protective immunity

against infectious diseases, as well as informing next generation vaccine and therapeutic development[1]. Deliberate inoculation of a well-defined virus to a screened population of participants results in a variety of infection outcomes unconfounded by inoculum dose, strain or differences in exposure, thus allowing predisposing immune factors associated with protection to be defined.

The SARS-CoV-2 human challenge study that we conducted in 2021–2022 demonstrated the safety and tolerability of such studies in healthy young volunteers[2]. 53% of inoculated seronegative participants became infected, while 47% resisted sustained infection but nevertheless showed evidence of local antiviral responses. Single cell transcriptomic analysis revealed innate and adaptive immune cell infiltration in the nasopharynx, early after inoculation in those who resisted sustained infection[3]. Transient infections (single non-consecutive PCR detections) resulted in an increased abundance of DCs, monocytes, macrophages, MAIT cells, T cells and NK cells in the nasopharynx at day 1 post-inoculation with few changes in the blood (PBMC) compartment. In contrast, the response of participants with abortive infections (PCR negative throughout) was limited to early increases in T cell abundance and activated conventional dendritic cells (cDC) only in the nasopharynx and activated MAIT cells in the blood[3]. Thus, mucosal immune responses during the pre-symptomatic period were associated with functionally complete protection. However, what pre-existing immune factors might underlie or contribute to this protective response remained unclear.

While responses during primary COVID-19 have been studied extensively, protective immune factors at the time of virus encounter are less well understood and almost impossible to accurately define during natural infection. Pre-existing cross-reactive antibodies induced by previous seasonal coronavirus infections may confer protection if present at sufficiently high levels but the literature is inconclusive about their importance[4]. T cells may play a role in asymptomatic infection or as early controllers of COVID-19, potentially terminating infection prior to development of antibodies or qPCR positivity[5–7]. This was suggested by studies showing that pre-existing non-structural protein (NSP)-specific T cell populations against the replication transcription complex (RTC) were expanded in highly exposed but uninfected healthcare workers, implicating cross-reactive T cells in protection against infection with SARS-CoV-2[8]. In individuals with no prior specific immunity, variations in innate immune function are likely to play a particularly important role in protection against infection. These variations may be due to previous pathogen exposure, for example, trained immunity or human cytomegalovirus (HCMV) driving differentiation of NK cells, or other environmental or genetic factors, but in humans these are relatively poorly understood.

Our previous analyses of the mucosal and systemic immune responses important for viral control showed not only the association between early mucosal responses and immediate clearance of infection, but also that CD8+ T cell activation and production of mucosal antibodies strongly correlated with the speed of later viral load decline[9]. Now, by performing detailed and comprehensive measures of pre-existing immunity followed by multi-variate and network analysis, we identify key innate and adaptive factors pre-exposure that predict functionally complete protection in seronegative adults with implications for future pandemic preparedness.

## Results

### Divergent outcomes in seronegative adults inoculated with SARS-CoV-2

Thirty-four seronegative healthy volunteers aged 18-29 were inoculated with 10 $TCID_{50}$ of SARS-CoV-2/human/GBR/484861/2020 (a D614G-containing pre-Alpha wild-type virus) on day 0. Nasal swabs and PBMC collected at day −1/−2/0 (baseline), day 1, 3, 5, 7, 10, 14 and 28 post-inoculation and plasma samples collected daily were used to assess the pre-exposure and early immune factors associated with

outcome (Fig. 1a). As described previously[2], 18 participants developed PCR+ and culture-confirmed infection as per the protocol definition of infection (at least 2 quantifiable qPCR detections on consecutive swabs), constituting the infected group. The remaining 16 did not meet the criteria for infection and were therefore grouped as the uninfected group (Fig. 1b).

Five of these uninfected participants, however, developed sporadic (single, non-consecutive) low-level qPCR detections (above the the lower limit of quantification (LLOQ), 3 $\log_{10}$ copies per ml) between 1.5- and 8.5-days post-inoculation in either the nose or the throat and were sub-grouped as transiently infected (Fig. 1b and Supplementary Fig. 1a). The remaining 11 uninfected participants had no qPCR detections above LLOQ (positive detections less than the LLOQ were assigned a value of 1.5 $\log_{10}$ copies per ml) and were sub-grouped as abortive infections due to evidence of local immune responses implying at least viral cellular entry[3]. The criteria for classification of transient and abortive infection were described previously[3].

Neither transient nor abortive groups met the protocol-defined criteria for sustained infection and re-analysis of previously-presented data[2,9] showed that transient infections, similar to the abortive infections, resulted in virtually no symptoms (Supplementary Fig. 1b), no seroconversion (Supplementary Fig. 1c) and no peripheral blood T cell responses as measured by flow cytometry (Supplementary Fig. 1d). In clinical practice, transient and abortive infections cannot therefore be readily detected or distinguished and represent equally favourable outcomes. For the purposes of this analysis, we therefore primarily focused on comparing the most clinically-relevant outcomes of sustained infection (protocol-defined) versus the combined uninfected group with functionally complete protection.

### Pre-existing cross-reactive antibody levels are poorly predictive of protection against infection but correlate with delayed onset of viral shedding

In the context of endemic respiratory viruses that cause re-infection throughout life such as influenza virus and RSV, protection correlates with circulating and/or mucosal antibodies[10]. Since the participants in this study were seronegative to SARS-CoV-2, we first sought to test the hypothesis that pre-existing levels of cross-reactive antibodies generated after earlier seasonal human coronavirus (hCoV) infections were associated with protection. IgM, IgG and IgA against hCoV 229E, HKU1, NL63 and OC34 spike, SARS-CoV-1 spike, and SARS-CoV-2 spike and nucleocapsid (N) were measured in plasma and nasal lining fluid collected at baseline (day -1; pre-inoculation) by Mesoscale Discovery (MSD). Overall, no significant differences in systemic (Fig. 2a) or mucosal (Fig. 2b) antibody levels were found between the 2 outcome groups prior to inoculation. However, 6 participants in the uninfected group were noted to have modestly higher levels of serum anti-SARS-CoV-2 N IgG at baseline (Fig. 2a).

Subgroup analysis showed these individuals falling within the abortive infection group, with anti-SARS-CoV-2 N IgG significantly higher (without multiple testing correction for antigen and isotype comparisons) in the abortive group compared with both transient ($p = 0.0332$) and infected groups ($p = 0.0131$) (Supplementary Fig. 2a). Additionally, the mucosal anti-SARS-CoV-2 S IgA antibody level was significantly higher in the abortive group compared with the transient group ($p = 0.0009$) (Supplementary Fig. 2b). Thus, protection against detectable viral replication was associated with pre-existing antibodies albeit based on small subgroups. These antibodies could have been derived from earlier seasonal coronavirus infection, although asymptomatic SARS-CoV-2 infection, without seroconversion cannot be fully ruled out.

In addition, the variability in delay between inoculation and qPCR positivity (time to qPCR positive) in the infected group was positively correlated with baseline levels of several nasal antibodies, primarily

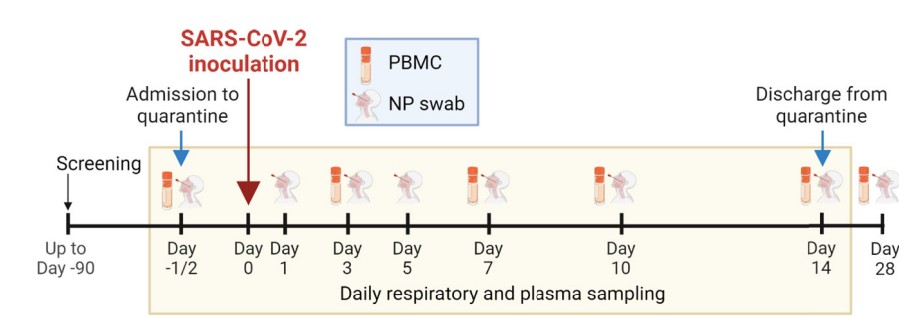

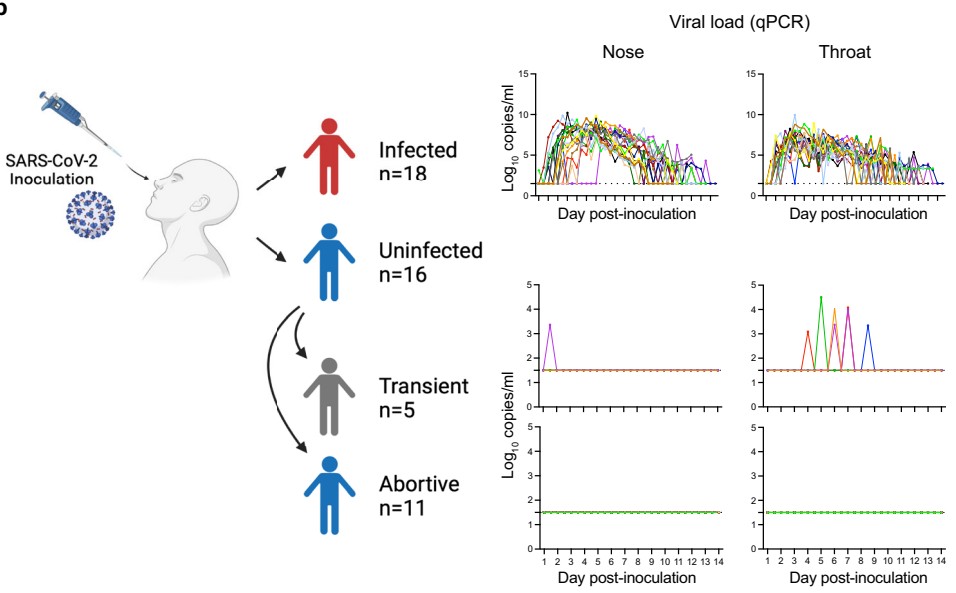

**Fig. 1 | Divergent outcomes in seronegative adults inoculated with SARS-CoV-2.**
**a** Schematic of the study timeline and sample collection timepoints utilised for this analysis. **b** Diagram of the infection outcome groups and subgroups, and the associated viral loads measured in daily nose and throat swabs by

qPCR ($\log_{10}$ N gene copies/ml); viral load data was published previously[2]. Images created in BioRender. Wagstaffe, H. (2025) https://BioRender.com/nqvesc4, https://BioRender.com/hh208ml.

IgM binding SARS-CoV-2 S, but there was no correlation with plasma antibody levels (Fig. 2c). Increased time to qPCR-positivity was thus associated with an overall shorter duration of qPCR positive infection (Fig. 2d), suggesting a potentially favourable relationship between baseline cross-reactive antibody levels in the mucosa and reduced viral shedding. Together, these data suggest that pre-existing cross-reactive antibodies at the levels found in these young adult volunteers might potentially have contributed to some protective effects but overall did not clearly predict protection.

### Protection from sustained SARS-CoV-2 infection is associated with elevated nasal chemokine levels

In the absence of high levels of virus-specific antibodies capable of preventing infection, other immune mechanisms potentially play a more important protective role. Having previously identified specific patterns of increased immune cellularity in the nasopharynx early after inoculation in protected individuals[3], we hypothesised that chemokine gradients present at the time of virus exposure might predispose to rapid local cellular recruitment. Thirty-five cytokines and chemokines in nasal lining fluid and plasma at the pre-inoculation timepoint were assayed by MSD; the full soluble mediator data set was presented previously[9]. At baseline, a higher protein level of CCL13 (monocyte chemotactic protein-4; MCP-4; $p = 0.0199$) was observed in the uninfected compared with the infected group (Fig. 3a), though this did not

survive p-value adjustment for the number of mediators measured (Supplementary Table 1). Baseline CCL22 (macrophage derived chemokine; MDC) was also higher in the uninfected compared with the infected group, but this was not statistically significant (infected; median 51.04 pg/ml vs uninfected; 69.96 pg/ml, corrected $p = 0.164$) (Fig. 3b). CCL13 and CCL22 act as myeloid and lymphoid cell chemoattractants and are produced by multiple cell types[11]. At day 2 post-inoculation, CCL13 protein remained significantly higher in the uninfected compared with the infected group (Supplementary Table 2, adjusted $p = 0.00159$) (Fig. 3c), while the level of CCL22 protein measured in the nasal lining fluid followed the same trend but was again non-significant (Fig. 3d).

To determine which cells expressed these chemokine genes most highly, we next analysed single cell RNA sequencing data from a subset of the cohort (16 participants; 6 infected, 3 transient, 7 abortive). CCL13 and CCL22 RNA were most evident in the myeloid cell compartment of the nasopharyngeal cells (tissue resident myeloid due to location) collected by swab (Supplementary Fig. 3a and Fig. 3e) and the monocyte and DC compartment of PBMC (Supplementary Fig. 3b). Expression of nasal CCL13 RNA in individual cell populations was low overall, but evident in DCs expressing CD69 and ITGAE (CD103) and monocytes at day -1 and 1 in the uninfected group. While lacking in the infected group at day -1 and 1, CCL13 was upregulated at day 7 and 10 in monocytes and cDCs (Fig. 3f). DCs expressing CD69 and ITGAE in the

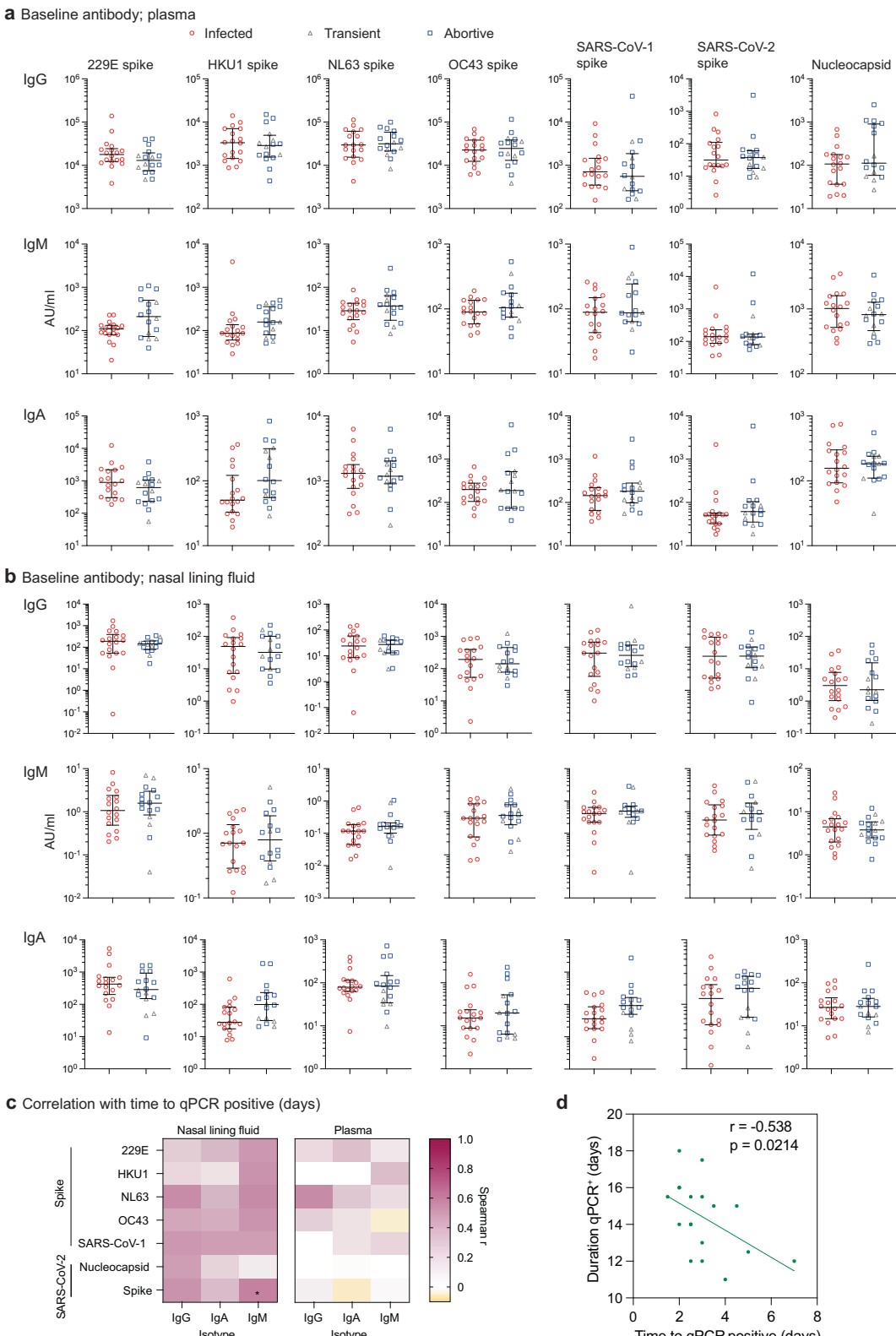

**Fig. 2 | Pre-existing cross-reactive antibody levels are poorly predictive of protection against infection but correlate with delayed onset of viral shedding.** **a** Plasma and **b** mucosal lining fluid IgG, IgM and IgA against hCoVs 229E, HKU1, NL63 and OC34 spike, SARS-CoV-1 spike and SARS-CoV-2 spike and nucleocapsid in the infected (*n* = 18) and uninfected (*n* = 16) groups measured at baseline (day −1) by MSD (transiently infected shown as grey triangles, *n* = 5). **c** Correlation matrix heatmap showing Spearman correlations between baseline antibody level and time to qPCR positivity in days corrected for multiple comparisons by false discovery rate (FDR). **d** Two-sided spearman correlation between time to qPCR positivity and duration of qPCR positivity in days. Median lines and IQR are shown. Two-sided multiple Mann–Whitney unpaired tests corrected for multiple antigen and isotype comparisons (Holm–Šídák's) was used to show significance between the two groups. *$p < 0.05$.

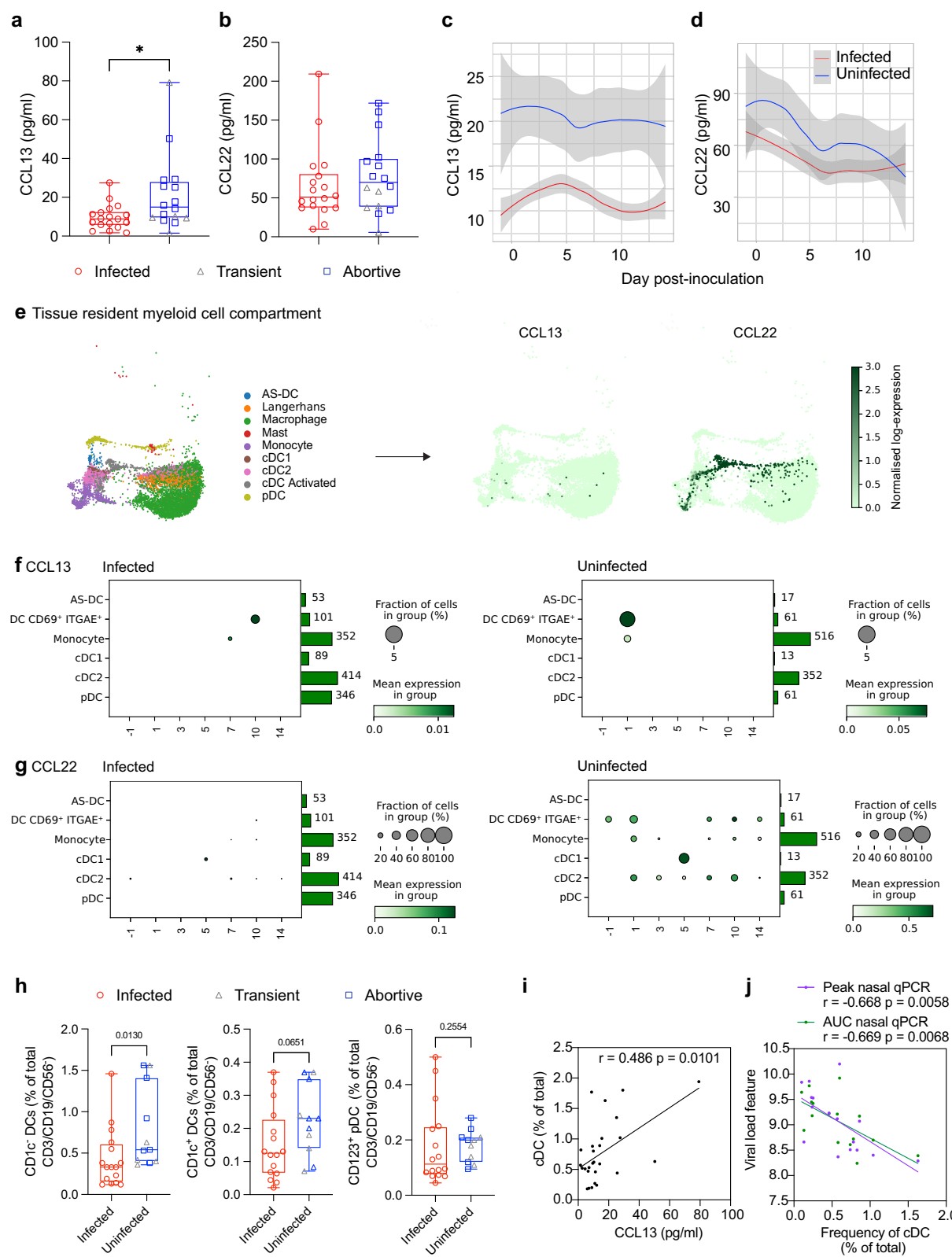

**f** CCL13

**g** CCL22

uninfected group also expressed CCL22 RNA at day -1 and across the time course, along with monocytes and cDC2 (Fig. 3g). We also showed RNA expression of CCR1 and CCR2, receptors for CCL13, in several circulating myeloid populations of the uninfected group, most notably plasmacytoid DCs (pDC), cDC and monocytes (Supplementary Fig. 3c).

To investigate further, we measured the frequency of circulating myeloid cell subsets by flow cytometry in whole blood. CD1c⁻ and

CD1c⁺ DC populations were gated and expressed as percentage of total CD3⁻CD19⁻CD56⁻, low SSC cells (gating strategy in Supplementary Fig. 3d). Pre-inoculation, there was a significantly higher frequency of CD1c⁻ DCs in the uninfected group compared with the infected group, while no significant difference was seen in CD1c⁺ DCs and pDCs (Fig. 3h). The frequency of total cDC in the blood correlated positively with the level of soluble CCL13 at baseline in the nasal lining fluid, these

**Fig. 3 | Protection from sustained SARS-CoV-2 infection is associated with elevated nasal chemokine levels. a** CCL13 and **b** CCL22 concentration in the nasal lining fluid of the infected ($n = 18$) and uninfected ($n = 16$) groups measured by MSD at baseline (transiently infected shown as grey triangles, $n = 5$). **c** CCL13 and **d** CCL22 concentration in daily nasal lining fluid samples pre- and post-inoculation in the infected ($n = 18$) and uninfected ($n = 16$) groups shown as loess plots. **e** UMAP of nasopharyngeal tissue resident (location) myeloid cell compartment expression of CCL13 and CCL22 RNA. **f** Nasopharyngeal myeloid cell CCL13 and **g** CCL22 RNA expression in the infected ($n = 6$) and uninfected ($n = 10$) groups pre and post-inoculation. Circles represent the fraction of cells expressing CCL13 or CCL22, circle colour indicates higher expression. Bars show the number of cells sequenced.

**h** DC populations in whole lysed blood were gated using CD1c and CD123. Plots show baseline frequencies in the infected ($n = 18$) and uninfected ($n = 16$) groups as a percentage of the CD3⁻CD19⁻CD56⁻ gate (gating strategy shown in Supplementary Fig. 3d) (transiently infected shown as grey triangles, $n = 5$). Spearman correlation between **i** the baseline total cDC frequency in blood and CCL13 concentration in the nasal lining fluid and **j** total cDC frequency in blood and viral load peak or AUC, no correction for multiple comparisons. Box and whisker plots show minimum to maximum, median, 25ᵗʰ and 75ᵗʰ percentile and all points. Unpaired tests of significance were undertaken by two-sided Mann–Whitney *U* test (**a**, **b**, **h**). *$p < 0.02$ or $p$ values shown.

data together suggesting that cDCs could be one of the main sources of CCL13 in the nasopharynx (Fig. 3i).

Post-inoculation, there was no change in the frequency of DC subsets in either uninfected or infected individuals (Supplementary Fig. 3e). However, pre-inoculation cDC frequencies negatively correlated with viral load in those who became infected (Fig. 3j). Our previous analysis demonstrated a strong correlation between activated CD8⁺ T cells and viral clearance[9], suggesting an indirect relationship between cDCs and viral clearance via antigen presentation. No significant differences in soluble CCL13, CCL22 or DC subset frequency were measured between the transiently infected group and the infected or abortive group (Supplementary Fig. 3f, g, h). In particular, while not powered to test differences between these subgroups, there was no evidence that CCL13, CCL22 or DC frequencies were elevated in transiently infected individuals, suggesting that these were more akin to the infected than the abortive group. Together, these data suggest that baseline myeloid and lymphocyte chemoattractant CCL13 and CCL22 levels in the nose, primarily produced by DCs, may be associated with prevention of infection.

IL-18 was also significantly different between infected and uninfected, but in contrast to CCL13, was higher in the infected compared with the uninfected at day 0 ($p = 0.0424$, Supplementary Table 1). This again did not survive p-value adjustment for multiple comparisons. Furthermore, it demonstrated a more erratic pattern of expression than CCL13 and CCL22, with the difference only seen at baseline timepoint day 0, not at day -1 (Supplementary Tables 1 and 3 respectively) and without the same sustained expression as CCL13 and CCL22 into the post-inoculation period[9] (Supplementary Fig. 3i). Given that the difference in IL-18 only just reached statistical significance and expression was highly variable, this was not investigated further.

### Less differentiated, NKG2C⁻ NK cells at the time of inoculation are associated with protection

Along with myeloid cell recruitment, single cell transcriptomics of nasal cells demonstrated increased NK cell abundance and cycling in the transient infection group at day 1 post-inoculation, suggesting a potential role in early viral containment[3]. As hCoV-specific antibodies pre-inoculation were not elevated overall in the protected participants, we hypothesised that non-antibody mediated NK cell functions were likely to be more important than Fc-mediated mechanisms such as antibody-dependent cellular cytotoxicity (ADCC). We therefore proceeded to examine NK cell phenotype and function at baseline and the early post-inoculation response in more detail.

NK cells of the nasopharynx were *NCAM1* low, *B3GAT1* negative, *FCGR3A* low, *KLRC1* positive and *KLRC2* low (CD56ˡᵒʷ, CD57⁻, CD16ˡᵒʷ, NKG2A⁺ and NKG2Cˡᵒʷ, respectively) (Supplementary Fig. 4a) primarily representing less differentiated, cytokine responsive/secreting NK cell populations[12]. To further characterise NK cells in this cohort, PBMC from a subset of participants at day -1, 3, 7 or 10, 14 and 28 (n numbers per timepoint per group is shown in the figure legend, with $n = 3$ HCMV seropositive in each group) were assessed for NK cell phenotypic markers and analysed by unsupervised analysis and manual gating (strategy in Supplementary Fig. 4b).

T-SNE and FlowSOM analysis of baseline samples ($n = 13$ infected, $n = 14$ uninfected) revealed clusters that differed between the infected and uninfected groups (Fig. 4a). Cluster 1 and 5 were significantly higher in count and frequency in the infected group compared with the uninfected group implying an association with susceptibility; no other clusters differed significantly in abundance (Fig. 4b and Supplementary Fig. 4c). FlowSOM cluster 1 displayed a CD56ᵈⁱᵐCD57⁺CD16⁺NKG2C⁺ phenotype and cluster 5, which was low or absent in the majority of uninfected participants, displayed a CD56ᵈⁱᵐCD57⁺CD16⁻NKG2C⁺ phenotype, both representing highly differentiated NK cell subsets[12,13] (Supplementary Fig. 4d). Furthermore, NK cells of cluster 1 in the infected group had higher CD57 and NKG2C MFI than those of the uninfected group and FcεR1γ⁻ (marker of adaptive) NK cells present in the infected group were low or absent in the uninfected group (Fig. 4c).

To confirm the presence of adaptive NK cells, manually gating of the adaptive (CD56ᵈⁱᵐCD57⁺FcεR1γ⁻) NK cell subset (gating strategy Supplementary Fig. 4b) was overlayed and localised to cluster 1 and 5 of the tSNE map (Fig. 4d, in green). Manual gating confirmed a higher frequency of less differentiated CD56ᵈⁱᵐCD57⁻ NK cells in the uninfected group compared with the infected group, with reciprocal higher frequencies of more differentiated CD56ᵈⁱᵐCD57⁺ in the infected group, which was significant at day 3 post-inoculation (Fig. 4e). This difference was upheld when the 3 infection groups were compared: there was a higher frequency of CD56ᵈⁱᵐCD57⁺ NK cells in the infected group compared with the abortive group at day 3 (Supplementary Fig. 4e). Regions in the baseline t-SNE map including regions A and B (marked with an arrow in Fig. 4f) that displayed a less differentiated phenotype (CD56ᵈⁱᵐCD57ⁱⁿᵗᵉʳᵐᵉᵈⁱᵃᵗᵉ/⁻CD16⁻NKG2Cˡᵒʷ/⁻) similar to those of the phenotype found in the nasopharynx (shown in Supplementary Fig. 4a) were enriched in the uninfected group (Fig. 4f) and the CD57:CD57⁺ ratio was higher in the uninfected group, although these were not statistically significant (Fig. 4g).

The 3 HCMV seropositive participants in each group were evenly distributed across the CD56ᵈⁱᵐ CD57⁺NKG2C⁺ and CD57⁺FcεR1γ⁻ frequency range suggesting HCMV infection (known to expand populations of highly differentiated and adaptive NK cells[13]), was not driving the differences observed between the two groups (Supplementary Fig. 4f). Together, these data suggest that enrichment of less differentiated, cytokine responsive/producing NK cells at baseline and early after inoculation, alongside a lower proportion of highly differentiated and adaptive NK cell populations (more specialised for ADCC responses), was associated with resisting infection. This is supported by single cell transcriptomic analysis demonstrating greater early activation and cytokine module scores in NK cells particularly at day 1 and 3 in the abortive infection group, contrasting with greater cytotoxicity module scores in the sustained infection group (Supplementary Fig 4g; genes used to distinguish modules listed in Supplementary Table 4).

To investigate the relationship between NK cell response and viral clearance, we next measured proliferation (indicated by upregulation of Ki-67) by flow cytometry. There was no evidence of proliferation in the uninfected or transient groups

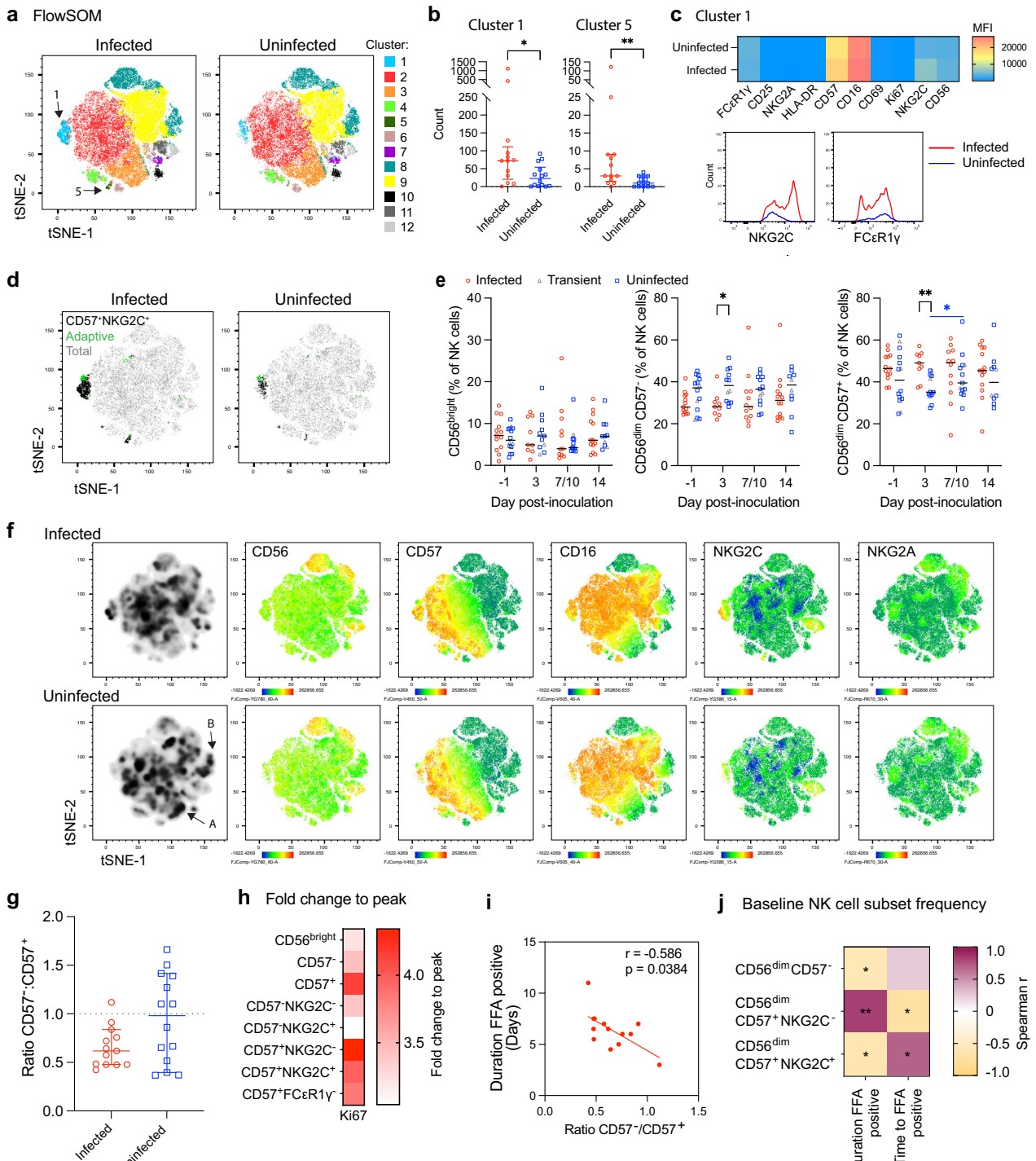

**Fig. 4 | NK cell NKG2C⁻ subsets at the time of inoculation are associated with protection.** The phenotype of NK cells in PBMCs was analysed by flow cytometry. **a** FlowSOM clustering of the baseline NK cell t-SNE map showing clusters 1–12, **b** cluster 1 and 5 count split by the infected ($n = 13$) and uninfected ($n = 14$) groups and **c** phenotype analysis of FlowSOM cluster 1. **d** Manually gated baseline CD56$^{dim}$CD57$^-$NKG2C$^+$ and adaptive (CD56$^{dim}$CD57$^+$FCeR1g$^-$) NK cells overlayed onto the t-SNE map (gating strategy shown in Supplementary Fig. 4b). **e** Manually gated NK cell subset frequencies determined by CD56 and CD57 at day −1, 3, 7 or 10 and 14 post-inoculation (baseline $n = 13$ infected, $n = 3$ transient, $n = 11$ abortive, day 3 $n = 9$, 2, 10, day 7/10 $n = 12$, 2, 11, day 14 $n = 14$, 2, 18, day 28 $n = 12$, 4, 11 respectively). **f** t-SNE plots showing CD56, CD57, CD16, NKG2C and NKG2A expression of PBMCs by heatmap. **g** The ratio of CD56$^{dim}$CD57$^-$ to CD56$^{dim}$CD57$^+$ NK cells, non-

significant by Mann−Whitney unpaired $U$ test (infected ($n = 13$) and uninfected ($n = 14$). **h** Heatmap of mean Ki67 expression fold change from baseline to peak (day 7 or 10) in the infected group split by NK cell subset. **i** Spearman correlation between the baseline ratio of CD57$^-$/CD57$^+$ and the duration of FFA positivity in days in the infected group. **j** Spearman correlation heat map between baseline NK cell subset frequency and duration FFA positive or time to FFA positive in the infected group, not corrected for multiple comparisons. Lines show median or median and IQR. Longitudinal analysis uses two-way ANOVA mixed-effects models with Geisser-Greenhouse correction and Tukey's or Šídák's (between groups) multiple comparisons test (**e**), 2 group analysis uses two-sided Mann−Whiney unpaired $U$ test (**b**, **g**). *$p < 0.05$, **$p < 0.01$, ***$p < 0.001$.

(Supplementary Fig. 5a), but in the infected group, all NK cell subsets upregulated Ki-67 with the highest peak frequencies in the CD56$^{dim}$CD57$^-$ subset (Supplementary Fig. 5b, gating strategy in Supplementary Fig. 5c). Greatest fold changes over baseline in the more differentiated CD57$^+$ and NKG2C$^+$ subsets (Fig. 4h) suggest robust responses by these cells occurred during infection, in line with their importance as antiviral effectors[14–16]. To further profile these cells, we tested the correlation between NK cell subset frequency at baseline and the course of infection in infected participants. A higher CD57$^-$:CD57$^+$ ratio at baseline significantly correlated with a shorter duration of focus forming assay (FFA; live virus) positivity (Fig. 4i). Higher CD56$^{dim}$CD57$^-$ and CD56$^{dim}$CD57$^+$NKG2C$^+$ NK cell frequency similarly correlated with a shorter duration of viral shedding and also with delayed time to FFA positivity (Fig. 4j). Conversely, a higher CD56$^{dim}$CD57$^+$NKG2C$^-$ NK cell frequency was associated with a longer duration of live virus positivity and shorter time to FFA positivity (Fig. 4j). Together, these data suggest that if sustained infection does occur, NKG2C$^+$ NK cells are associated with favorable outcomes.

### Non-structural protein-specific, IL-2 producing T cells are present in seronegative individuals and enriched in those who remain uninfected

In those who remained uninfected, single cell RNA had previously revealed an early increase in the abundance of infiltrating and tissue resident CD4$^+$ and CD8$^+$ T cells in the nasopharynx at day 1 post-inoculation (Fig. 5a; reanalysed from[3]). This was distinct from those who developed sustained infection, where infiltrating T cells were predominantly found at 5−14 days post-inoculation. The rapidity of this early response suggested the presence of pre-existing memory T cells associated with resistance to infection. Measurement of baseline peripheral SARS-CoV-2 spike-, nucleocapsid- and membrane-specific T cell responses by ELISpot showed frequencies below the LLOQ and no differences in IFN-γ producing T cells between the groups (Supplementary Fig. 6a). We therefore sought to measure the peripheral T cell response to antigens of the replication transcription complex (RTC), which are well-conserved across hCoVs and have been previously shown to be expanded in highly-exposed healthcare workers who remained seronegative to SARS-CoV-2[8].

Baseline PBMC from a subset of participants (infected ($n = 13$), transient ($n = 5$) and abortive ($n = 10$)) were stimulated with peptide pools of the RTC (NSP7, 12 and 13) of SARS-CoV-2 and IL-2 and IFN-γ producing T cells were measured by Fluorospot (Fig. 5b). The majority of participants in the infected and uninfected groups responded to peptide pools covering NSP7, 12 and 13, despite no serological evidence of prior COVID-19 infection (Fig. 5c). Although non-significant, IL-2 producing T cells tended to be enriched in the uninfected compared with the infected group, with a higher responder rate, although the small sample size limited the power of this analysis (Fig. 5c). IFN-γ and dual producing IL-2 and IFN-γ T cell responses were also detected in both groups at low frequency (Fig. 5c). T cell responses to RTC peptide stimulation were elevated at day 14 following sustained infection compared to day 28, but no increase over time was observed in the uninfected group (Fig. 5d). Baseline IFN-γ producing T cell frequencies against NSP12 in the uninfected group correlated positively with the pre-inoculation levels of HKU1 and OC43 anti-spike antibody, suggesting these cells may have been generated following previous seasonal beta-hCoV infections (Fig. 5e).

Separating the transient from abortively infected individuals suggest the pre-existing RTC-specific IL-2 producing T cell response was similarly enriched in both sub-groups, although underpowered for this comparison (Supplementary Fig 6b). Interestingly, the day 1 increase in nasopharyngeal T cell abundance was predominantly infiltrating CD4$^+$ T cells (CD103 and *ITGAE* negative) in the transiently infected subgroup; whereas, tissue resident CD103$^+$ CD8$^+$ T cells

dominated in the abortive subgroup (Supplementary Fig 6c). We speculate that RTC-specific T cells present prior to inoculation are involved in resisting detectable infection, and that CD8$^+$ resident memory T cells at the site of infection might be particularly important for the earliest termination of infection.

Since almost all participants had some degree of pre-existing RTC-specific T cell memory, we then tested the relationship between abundance of RTC-specific T cells and the course of infection. Baseline IFN-γ producing T cells in response to NSP12 peptide stimulation correlated positively with the magnitude (AUC) of the polyclonal activated and proliferating CD8$^+$ T cell response (Fig. 5f), the kinetics of which were reported previously[9]. Baseline NSP12 reactive IFN-γ producing T cells also negatively correlated with the duration of qPCR positivity (Fig. 5g). These findings therefore provide further support for the importance of CD8$^+$ T cells in viral clearance, with pre-existing cross-reactive T cells likely derived from earlier seasonal hCoV infections predisposing to more favourable outcomes against primary infection with the pandemic virus[9]. However, in the absence of high levels of virus-specific immunity, our data also show the importance of a coordinated, multi-factorial immune response and indicate how chemokines, DCs and NK cell subsets, antibodies and local T cells present at the time of virus exposure may all contribute to prevent and modulate viral replication early post-infection.

### Integrative machine learning identifies CCL13-CD1c$^+$ DCs as a protective axis

The antiviral immune response is highly coordinated and multiple arms of immunity are tightly co-regulated. It is therefore unsurprising that multiple immune factors (antibodies, soluble mediators, immune cells) were all associated with divergent clinical outcome. To assess the relative importance of these and infer their relationship with each other, we undertook multivariate analyses, initially training a logistic-regression classifier (with elastic-net regularisation) using baseline soluble mediators as predictors, these being the most complete and comprehensive dataset. Overall, the model showed good and stable discrimination of the uninfected versus infected groups across repeated cross-validation runs (AUROC 0.785; Fig. 6a, Supplementary Fig. 7a, b). However, few nasal proteins ranked by selection frequency showed strong statistical significance (Fig. 6b). CCL13 was among the consistently selected top important features but did not surpass the permutation-based significance threshold. Instead, IL−10 emerged as the top-rank feature by selection frequency (permutation-based $p = 0.089$) and was significant in its coefficient fit (Fig. 6b). Univariate analysis showed IL-10 to be higher in the infected group, potentially marking susceptibility to infection, although the difference was not significant by Mann−Whitney U test (Supplementary Fig. 7c). Overall, these results suggest that while the aggregate protein signature provides discriminatory information, individual feature contributions remain difficult to resolve once inter-feature dependencies are considered.

Next, we aimed to delineate the dependency (and potential regulatory) structure among baseline features, several of which were associated with protection. Using machine learning, we inferred a conditional independence network to identify associations between features that persist after statistically conditioning on all other features. Notably, the network recovered the dependency between CCL13 and circulating CD1c$^+$ DC frequency (Fig. 6c, Supplementary Fig. 7d), further supporting CCL13-DC as a protective axis. There was also a relationship between CCL13 or CCL22 and NK cell CD57 ratio, suggesting a role for these chemokines in NK cell composition. In addition, the network revealed a strong association between circulating CD1c$^+$ DC frequency and the NSP-specific IL-2 producing T cell response (Spearman correlation $r = 0.620$, $p = 0.006$; Supplementary Fig. 7d), which further supports the involvement of CD1c$^+$ DCs in T cell activation through antigen presentation and co-stimulation. In contrast, IL

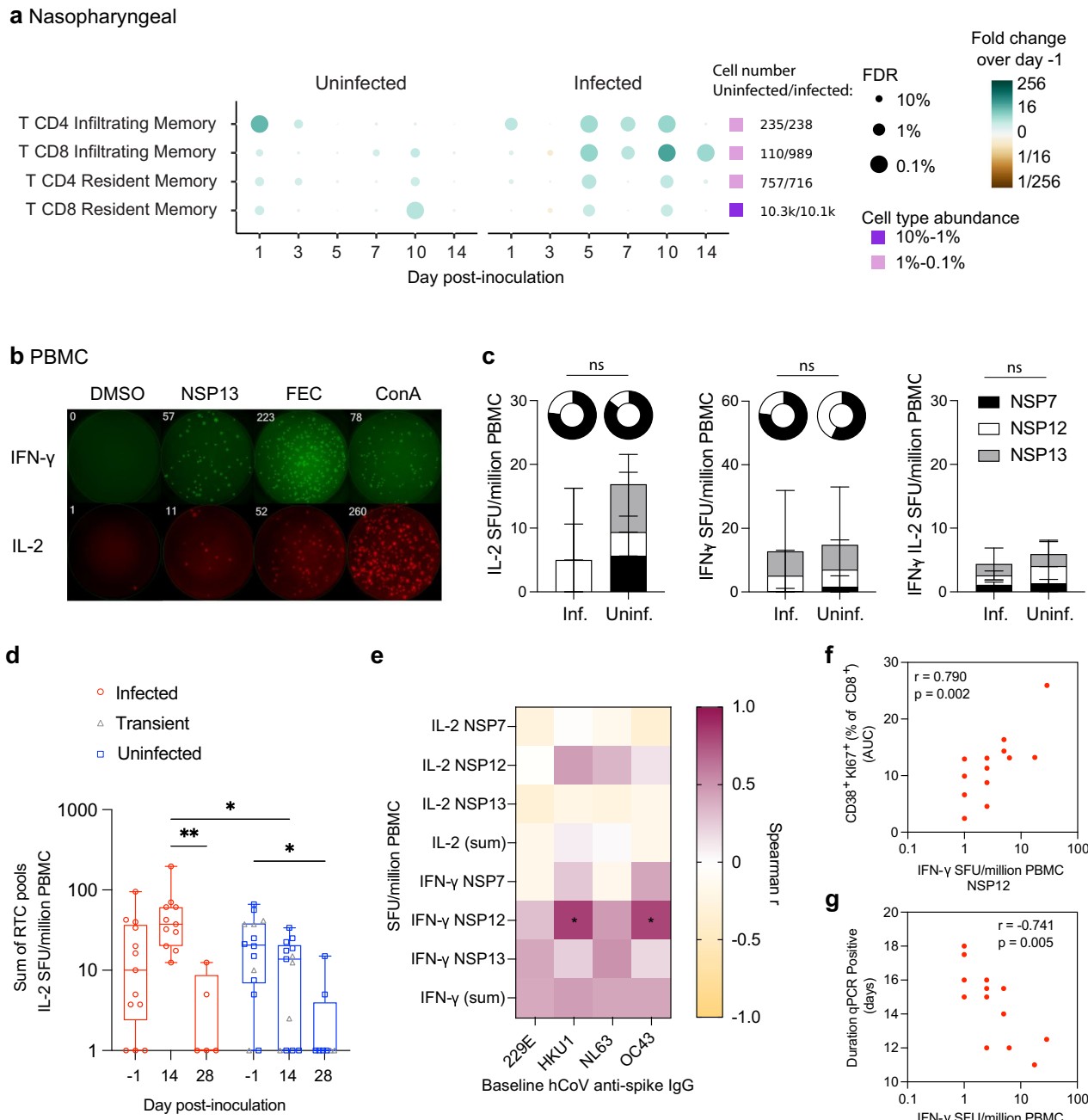

**Fig. 5 | Non-structural protein-specific, IL-2 producing T cells are present in seronegative individuals and enriched in those who remain uninfected.**
**a** scRNA sequencing showing abundance of infiltrating T cells and tissue resident T cells in the nasopharynx by fold change over baseline (day −1) in the infected (*n* = 6) and uninfected (*n* = 10) groups. **b** Representative Fluorospot data showing DMSO (negative control), NSP13 peptide pool, FEC (influenza, Epstein−Barr virus and CMV) and ConA (positive control) stimulated wells. **c** Stacked graphs showing baseline IL-2, IFN-γ and double IL-2 and IFN-γ spot forming units (SFU) upon NSP7, NSP12, NSP13 peptide stimulation measured by Fluorospot, and pie graphs show number of responders (responding to any one of NSP7, NSP12, NSP13 peptide pool stimulation) in the infected (*n* = 13) and uninfected (*n* = 15) groups. **d** Sum of NSP7,

NSP12, NSP13 IL-2 SFU at baseline (*n* = 11 infected, *n* = 5 transient, *n* = 9 abortive), day 14 (*n* = 11 infected, *n* = 4 transient, *n* = 9 abortive) and 28 (*n* = 5 infected, *n* = 2 transient, *n* = 6 abortive) post-inoculation. **e** Spearman correlation matrix heatmap showing correlations between baseline anti-Spike IgG against seasonal HCoVs and baseline IL-2 or IFN-γ SFU of the uninfected group. Spearman correlation between **f** baseline IFN-γ SFU in response to NSP12 and the AUC of the CD8⁺ CD38⁺Ki67⁺ frequency in the infected group or **g** the duration of qPCR positivity in days. Box and whisker plots show minimum to maximum, median, 25th and 75th percentile and all points. Significance testing was done by two-sided Mann−Whitney unpaired *U* test (**c**) or one-way ANOVA mixed-effects models with Geisser-Greenhouse correction and Holm-Šídák's multiple comparisons test (**d**). *$p < 0.05$, **$p < 0.01$.

−18 and IL-10, which had emerged as potential markers of susceptibility did not form part of this cluster. Together, these analyses suggest a CCL13-CD1c⁺ DC-centered program at baseline that couples chemokine to T cell readiness, potentially explaining protection against primary viral infection.

## Discussion

Despite lacking specific immunity, not everyone becomes infected when encountering a novel pathogen. Here, we identified immune features associated with resistance to primary infection by studying participants who were exposed via inoculation to SARS-CoV-2 but

 9

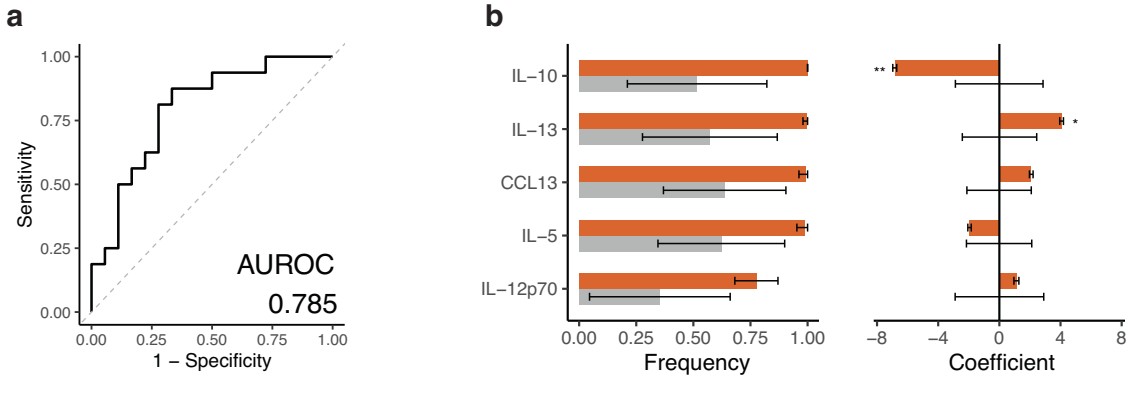

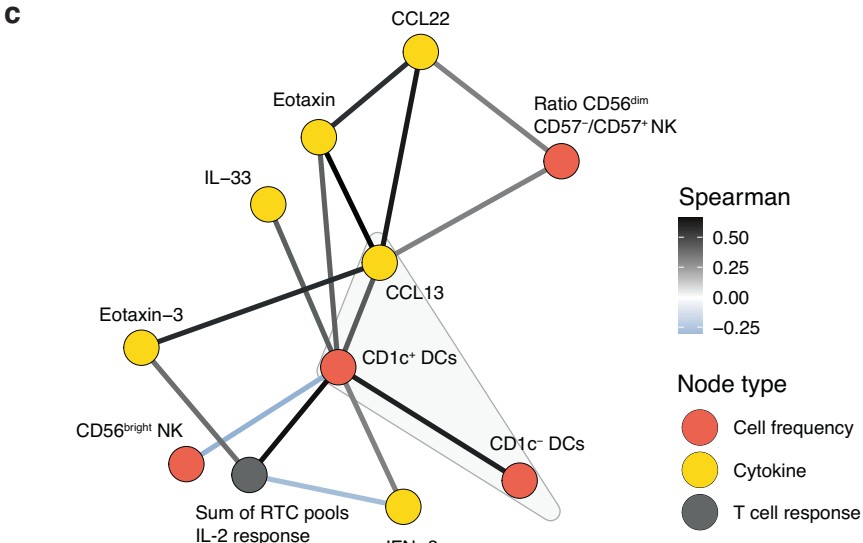

**Fig. 6 | Conditional independence network analysis reveals the dependency structure of baseline immune factors. a** Receiver Operating Characteristic (ROC) curve of classifying the uninfected participants ($n = 16$) versus those with sustained infection ($n = 18$) using regularised logistic regression across repeated cross validation (CV) runs ($n = 100$). The diagonal grey dashed line represents the performance of a random binary classifier. **b** Top predictive features ranked by selection frequency across CV runs ($n = 100$). Error bars represent the mean +/− standard deviation of feature selection frequencies and coefficients obtained from 100 randomly partitioned train-test datasets (orange), and from 100 null models per CV run by permutation of response labels (grey). Empirical one-sided $p$-values were computed by comparing observed values with permutation-based null distributions. *$p < 0.05$, **$p < 0.01$. **c** Conditional independence sub-network of baseline immune factors centered on 1023 the CCL13-DC axis (grey convex hull). Edge color reflects Spearman correlation coefficient.

remained uninfected. Of the pre-existing immune factors measured, pre-exposure CCL13 levels in the nasal mucosa correlating with cDCs were found to be key features associated with functionally complete protection. Additionally, pre-existing cross-reactive T cells and less differentiated NK cell subsets were higher in those who resisted infection, with network analysis demonstrating the dependent relationship between all these features. In contrast, low levels of antibodies in blood and nose, presumably generated by previous seasonal coronavirus infection showed modest association with protection. These results thus define the complex, multi-layered nature of immune protection in the context of emergent viruses with innate and early adaptive cellular immunity in the respiratory mucosa playing an especially important protective role.

SARS-CoV-2 human challenge of seronegative adults resulted in three different infection outcomes, but outside of the study setting, transient and abortive infections are inevitably grouped together as uninfected since they result in no clinical sequelae and cannot be readily diagnosed. Nevertheless, single-cell RNA sequencing had previously shown that none of these participants truly had sterilising immunity, with evidence of cellular immune responses at the mucosal

level in both transient and abortive subgroups[3]. With this classification, there was no clear relationship between pre-inoculation antibody levels and the uninfected outcome, in line with other studies showing antibodies from previous coronavirus infections not being a major player in protection against SARS-CoV-2[4]. Here, we additionally observed a sub-category of participants in the abortive group with moderately higher baseline systemic anti-N IgG and nasal anti-S IgA, which might have further contributed to the almost complete protection observed. Since baseline anti-S IgG or SARS-CoV-2-specific T cell frequencies were not elevated in these participants compared with others in the uninfected group, it is unlikely that these antibodies were due to earlier SARS-CoV-2 infection, although asymptomatic SARS-CoV-2 exposure cannot be fully ruled out. Instead, the sequence similarity of N between SARS-CoV-2 and other beta hCoVs, notably OC43, suggest that these could have been induced by recent seasonal coronavirus infections[17,18]. The importance of this is unclear, as these individuals were a small minority and antibody levels did not explain why others in the abortive or transient infection groups without such antibodies also did not develop sustained infection. It is theoretically possible that antibodies against other internal proteins that we did not

measure could constitute a further explanatory variable. However, cross-reactive antibodies against other internal proteins such as the NSPs of ORF1ab, while previously described following SARS-CoV-2 infection, are rare[19] with these proteins not part of the virion and only present when transcribed.

In contrast, our data suggest that the chemokine CCL13 (also known as Monocyte chemoattractant protein-4 [MCP-4]), which acts as a myeloid cell and lymphocyte chemoattractant, could be a more important factor in protection against primary infection. CCL13 is produced by multiple cell types, including structural, epithelial and myeloid cells, and binds several different chemokine receptors CCR1, 2, 3, 4, 5 and 11 to mediate its function[11]. Its role is best understood in asthma and other airway diseases, where it promotes recruitment of macrophages and eosinophils, increasing inflammation, but its involvement in respiratory viral pathogenesis is unclear. In children with viral exacerbation of asthma, CCL13 is upregulated during symptomatic disease and has been associated with macrophage recruitment[20]. One study has also shown higher levels in patients hospitalised with respiratory viral infection but no clear link to severity or protection has previously been demonstrated[21,22]. Furthermore, since there is no direct murine homologue and only partial functional overlap with CCL12 in the mouse, mechanistic investigation remains limited[23].

Here, we showed not only that nasopharyngeal DCs were expressing CCL13 but also a strong correlation between circulating DCs and nasal CCL13, further confirmed by network analysis. While the mechanism underlying this correlation could not be tested in this system, the finding suggests that there might be mutual reinforcement between local chemokines and DCs that traffic between blood and tissue with CCL13 a key mediator that represents a resting mucosal environment predisposed to the rapid cellular response upon inoculation associated with resisting infection. In our study, the reason for these differences in the pre-existing nasal environment is unclear, although recent respiratory infections, vaccinations or environmental factors may influence this. For example, up to 8 months following SARS-CoV-2 mRNA vaccination (BNT162b2), myeloid associated cytokine production capacity was enhanced in response to stimulation with bacterial TLR stimulants[24]. However, many of the uninfected group participants developed SARS-CoV-2 infection in the community during the follow-up period of the study, implying that this protective state might only be transient. Nevertheless, these findings suggest that sustained boosting of CCL13 specifically at the site of infection, perhaps by mucosal vaccination or other local immune-stimulating interventions, may be beneficial.

cDCs are essential for anti-SARS-COV-2 CD8$^+$ T cell responses in the lung, with an absence of cDC1 resulting in higher viral loads in a mouse infection model[25]. Network analysis of our data similarly revealed the dependent relationship between CD1c$^+$ DCs and NSP-specific T cells. In this setting, an early mucosal tissue resident T cell signature, that we speculate was cross-reactive for SARS-CoV-2, and higher frequencies of circulating T cells responding to RTC antigens were shown in the uninfected group. In these participants, no systemic adaptive immune responses were detectable, implying that early protective T cell responses had been entirely restricted to the respiratory tract. T cells against early viral proteins (translated first during viral replication, before a full replication cycle is complete[26,27]) have been proposed as mediators of complete protection if able to terminate the infection before spreading of the virus triggers systemic immunity and seroconversion. RTC proteins are expressed and presented earlier than structural proteins and contribute to more HLA-I presentation, although at lower levels, than structural proteins[28]. Clonally expanded TCRs that recognize NSP12 in SARS-CoV-2 unexposed individuals were able to kill a K562 cell line expressing NSP12 in vitro[29]. An earlier study in highly exposed healthcare workers had shown expanded populations of these cells in those who remained seronegative but with evidence of virus exposure by interferon-

stimulated gene upregulation[8]. These cells were also detected in pre-pandemic bronchoalveolar lavage samples, proving their cross-reactivity and generation by previous seasonal hCoV infection[30]. We previously showed that tissue resident CD8$^+$ T cells in the lung following controlled human influenza infection possessed innate-like properties, supporting their potential role in immediate protective responsiveness[31]. Thus, we propose that in the context of a coordinated immune response, cross-reactive T cells are likely to play an important role in protection and disease reduction, and our findings strongly support these as a target for cross-protective vaccination.

NK cells also play a critical role in viral control during infection, killing infected cells via multiple direct or indirect mechanisms, while producing cytokines to orchestrate the immune response within hours of viral exposure. IFN-γ$^+$ NK cells recruited to the lung after mucosal vaccination in a mouse model played a critical role in lung DC recruitment and activation and CD8$^+$ T cell responses[32]. Furthermore, depletion of NK cells in a non-human primate model of SARS-CoV-2 infection resulted in higher viral loads and a longer duration of infection[33]. The human NK cell subsets CD56$^{bright}$, CD56$^{dim}$CD57$^-$, CD56$^{dim}$CD57$^+$ and subpopulations of highly differentiated CD57$^+$NKG2C$^+$ and adaptive CD57$^+$FCεR1γ$^-$ subsets hold a diverse array of functions[12,13,34]. Less differentiated (CD56$^{bright}$ and CD56$^{dim}$CD57$^-$) NK cells are more responsive to cytokine stimulation than their more differentiated adaptive counterparts (CD56$^{dim}$CD57$^+$NKG2C$^+$, FCεR1γ$^-$), which preferentially mediate cytotoxic and antibody-dependent functions. Here, we showed higher frequencies of less differentiated NK cells in those who resisted infection, supporting the hypothesis that participants with a higher proportion of these less differentiated NK cells might be more able to clear virus early during primary infection with their greater cytokine responsiveness in the absence of strain-specific antibodies.

Adaptive NK cell subsets have been found to be expanded in the blood of patients with severe SARS-CoV-2 infections and are able to perform ADCC and secrete IFN-γ in response to SARS-CoV-2 peptides[14,15,35]. In this context, they are thought to be beneficial in the outcome of severe infection[14,15,35]. Furthermore, deletion or mutation of the NKG2C gene has been associated with severe disease, suggesting NKG2C$^+$ NK cells and their function are important in preventing progression to severe infection[15,36,37]. In the mild infections seen here, more highly differentiated NKG2C$^+$ NK cells were indeed associated with a delayed onset of infection and a shorter duration of live virus positivity. Thus, NK cell subsets may differentially impact primary infection with SARS-CoV-2, preferentially enhancing early and late viral clearance depending on their relative frequency and functions. In support of this, Strauss-Albee et al. demonstrated a greater NK cell diversity (shown to increase with antigen exposure) was associated with a higher risk of HIV-1 acquisition, potentially due to a decreased adaptability to respond to new pathogens[38]. This highlights NK cells and their diverse phenotypes and functions as an understudied measure of infection outcome and target for prevention.

The complex nature of protective immunity and inherent limitations of human studies make understanding of the multiple interconnected factors associated with differential outcome a challenge. However, machine learning enabled us to delineate a regulatory structure of a cluster of baseline features that we propose could be a key protective pathway against primary infection. In this hypothetical framework, CCL13, produced by resident DCs and other myeloid cell subsets in the nose, acts as a core mediator that readies the respiratory mucosa for rapid recruitment of DC subsets, NK cells and/or other innate cells from the blood. Upon viral infection, these cells then directly or indirectly through enhanced activation of Trm cells terminate viral replication before triggering any systemic immune responses.

The major limitations of this study are the low participant number per group, the young age range and early SARS-CoV-2 variant used for

challenge, which could constrain the generalisability of the conclusions drawn. Direct comparison of these findings in human challenge studies of previously infected and vaccinated cohorts will also be important, the SARS-CoV-2 immunological landscape having changed dramatically over time, with studies clearly showing a shift towards spike-specific and neutralising antibody-mediated protection in the seropositive context[39,40]. Nevertheless, these results signpost the immune pathways on which larger cohorts and further studies should focus for targets as correlates of protection for pandemic preparedness and to understand the inter-relatedness of immune mechanisms that might require more complex multi-variate correlates to be developed. These results argue for the targeting of local immunity for novel preventative measures in preparation for future outbreaks and pandemics, including topical immunostimulatory interventions, adjuvants and broadly protective mucosal vaccines.

## Methods

### Study design, ethical approval and participants

The study design, public consultation, and ethical approvals for the study have been detailed previously. The study was conducted in accordance with the protocol; the consensus ethical principles derived from international guidelines, including the Declaration of Helsinki and Council for International Organizations of Medical Sciences International Ethical Guidelines; applicable ICH Good Clinical Practice guidelines; and applicable laws and regulations. The screening protocol and main study were approved by the UK Health Research Authority's Ad Hoc Specialist Ethics Committee (references 20/UK/2001 and 20/UK/0002) and registered with ClinicalTrials.gov (identifier NCT04865237). Written informed consent was obtained from all volunteers before screening and study enrolment.

Thirty-four volunteers were healthy adults aged 18–29 years with no previous SARS-CoV-2 infection or vaccination (negative for SARS-CoV-2 anti-S antibodies measured at screening by MosaiQ COVID−19 antibody microarray (Quotient)). Participants were inoculated intranasally by pipette with 10 TCID$_{50}$ of wild-type SARS-CoV-2 (20 A clade of the B.1 lineage possessing the D614G mutation; GenBank accession number OM294022) between both nostrils (100 µl each). Safety was assessed with daily blood tests, spirometry, electrocardiograms, clinical assessments (vital signs, symptom diaries and clinical examination) and CT scan of chest on day 5 (all participants) and day 10 (infected participants only), safety and tolerability data were presented previously[2]. Eight participants (24%) were positive for HCMV serology tested at baseline in serum using Human Anti-CMV IgG ELISA (Abcam).

### Sample and data collection

Venous blood plasma (EDTA) and nasosorption samples were collected at day -1 and day 0 and daily post-inoculation and nasopharyngeal swab samples were collected at day -1, 1, 2, 5, 7, 10 and 14. Venous blood (lithium heparin) samples were collected at day -1, 3, 7, 10, 14, and 28 for PBMC isolation (see below). Nasosorption samples were eluted in 330 µl of immunoassay buffer AB-33k (Millipore) containing 1% Triton-X100 (v/v) (Sigma). Nasopharyngeal swab samples were cryopreserved in FCS/10% DMSO prior to processing for sequencing. PBMCs were isolated by density centrifugation using Histopaque 1077 (Sigma-Aldrich) as previously described[9]. Briefly, whole blood samples were diluted (1:1) in PBS and overlayered onto Histopaque, centrifuged for 30 min at 400 × g. Isolated PBMCs were washed in PBS and used immediately or cryopreserved in FCS/10% DMSO.

Viral load was measured from mid-turbinate and throat flocked swabs placed in 3 ml of viral transport medium (BSV-VTM-001, Bio-Serv) that was analysed by RT-PCR and quantitative culture by FFA as previously described[2]. Symptom diaries were self-completed three times daily from day -1 to day 14 after inoculation, serum anti-spike antibody level was measured by ELISA (Nexelis). Viral load, symptoms and antibody levels were reported in detail previously[2,9].

### Quantification of soluble mediators and antibody by MesoScale Discovery

The full data set of antibody and soluble mediator levels was presented in detail previously[9]. A panel of 35 cytokine and chemokine immune mediators were measured by MesoScale Discovery (MSD) multiplex immunoassays, consisting of CCL2, CCL3, CCL4, CCL13, CCL22, CXCL10, Eotaxin, Eotaxin-3, GM-CSF, IFNα2a, IFNβ, IFN-γ, IFN-λ (IL-29) IL-1α, IL-1β, IL-2, IL-4, IL-5, IL-6, IL-7, IL-8 (CXCL8), IL-10, IL-12p40, IL-12p70, IL-13, IL-15, IL-16, IL-17A, IL-18, IL-33, TARC, TNF, LTα, TSLP and VEGF-A per manufacturer's instructions. Unquantifiable samples were given a value of the lower limit of detection, denoted on plots as a dashed line. SARS-CoV-2 and hCoV anti-Spike antibody titers were similarly quantified using MSD multiplex immunoassay Coronavirus panel 2. Antibody panels were developed using anti-human IgG, IgM, or IgA and binding titres given as arbitrary units per milliliter (AU/ml) based on a kit-provided human standard curve.

### Flow cytometry

For myeloid cell staining, fresh whole blood was RBC lysed using Pharmlyse (BDbiosciences), washed in PBS then stained Zombie UV Fixable Viability Kit (Biolegend) according to manufacturers instructions. Cells were then stained for CD3 BV510, CD123 APC, CD56 BV510 (all Biolegend), CD14 APC-H7, CD19 BV510, HLA-DR PE-CF594 and CD1c APC-R700 (all BDbiosciences) in Brilliant Stain Buffer (BDbiosciences) for 30 min at room temperature. Cells were washed in PBS with 2% FBS then fixed with CellFix (BDbiosciences) for 20 min at 4 °C, washed and resuspended in PBS 2% FCS. T cell CD38 and Ki-67 expression used for correlation analysis was obtained by staining freshly isolated PBMC samples, methods are detailed previously[9]. Data were acquired on a BD LSRFortessa.

For NK cell staining, cryopreserved PBMC were thawed into pre-warmed RPMI with 10% FCS and washed and strained through a 40 mm cell strainer. A subset of participants were used based on cell availability (numbers are stated in the figure legends), either day 7 or day 10 was used. There were three participants that were HCMV seropositive in each infected and uninfected groups. Cells were counted using a Countess Automated Cell Counter (Invitrogen) and rested for a maximum of 2 h. Cells were stained with Zombie UV Fixable Viability Kit (Biolegend) according to manufacturers instructions, followed by surface staining with CD57 e450, CD25 PerCP Cy5.5 (Invitrogen), CD56 PE-Cy7, HLA-DR APC R700 (BDbiosciences), CD3 BV510, CD69 BV711, NKG2C PE, NKG2A APC, and CD16 BV605 (all Biolegend) in Brilliant Stain Buffer. Cells were washed in FACS buffer (PBS containing 2% FCS and 2 mM EDTA) then fixed and permeabilised for intranuclear staining using Foxp3/Transcription Factor Staining Buffer Set (ThermoFisher). Ki67 BV786 (BDbiosciences) and FCεR1γ FITC (Merck) were then added for 30 min at 4 °C. Data were acquired on a BD FACSymphony. Antibody details are in Supplementary Table 5.

### Single cell RNA transcriptomics

scRNA transcriptomics was performed on a subset of participants, $n = 6$ infected, $n = 3$ transient and $n = 7$ uninfected, detailed methods are described previously[3]. In brief, processed single cell RNA-sequencing data (accessible through the COVID-19 Cell Atlas web portal: https://covid19cellatlas.org; raw sequencing data deposited under controlled access at the European Genome-Phenome Archive under accession number EGAD00001012227[3]) was analysed using the single-cell analysis Python workflow Scanpy (Cite: PMID: 29409532). The original cell type annotations were used for all downstream analysis. Fold changes in nasopharyngeal immune cells over time were modeled using generalised linear mixed models (GLMMs) with a Poisson outcome (Cite: PMID: 38898278). Participant identifiers and days since inoculation were

included as random effect terms. Models were fit on relative immune cell abundances in each sample using the glmer function of the lme4 package implemented in R. Fold changes for each factor were estimated using the ranef function of the lme4 package and standard errors of the variance were calculated with the numDeriv package. Log-transformed fold changes are relative to the pre-inoculation time point (day-1). The local true sign rate and Benjamini-Hochberg procedure were used to determine false-discovery rates (FDRs) to assess the statistical significance of the fold change estimates.

## T cell ELISpot

Ex vivo ELISpot was carried out on fresh PBMC samples by Oxford Immunotec and data was presented previously[9]. Briefly, T-Cell Xtend reagent (Oxford Immunotec Abingdon, UK) was added to density gradient isolated PBMC to extend survival, PBMCs were counted using flow cytometry to ensure an adequate number in all patient's samples. IFN-γ secreting T cells specific to spike (N and C terminus), nucleocapsid and membrane protein specific antigen panels based on the Wuhan-Hu-1 sequence (YP_009724390.1) were detected using the T-SPOT Discovery test performed, according to the manufacturer's protocol. Antigen-specific T cell frequencies were reported as spot forming cells (SFC) per 250,000 PBMCs.

## T cell Fluorospot

Cryopreserved PBMC were thawed and washed in CTL Anti-Aggregate Wash, resuspended in CTL Serum-Free Media (all ImmunoSpot) and rested for a maximum of 2 h. A subset of participants were used based on vial availability (numbers are noted in the figure legends). Cells were counted using a Countess Automated Cell Counter (Invitrogen). Peptides were diluted in CTL Serum-Free Media and added to pre-coated 3-colour (IFN-γ, TNF and IL-2) Fluorospot plates (all Immunospot) at 1 µg/ml. SARS-CoV-2 peptides used to stimulate PBMC were previously described[8], NSP7 (15 peptides), NSP12 (36–37 per pool, 5 pools combined into a single well) and NSP13 (39–40 peptides per pool, 3 pools combined into a single well). PBMC plus anti-CD28 co-stimulation were added to each well at 400,000 per well and incubated for 18 h at 37 °C with 5% $CO_2$. Internal plate controls were two DMSO wells (negative controls), concanavalin A (ConA; Sigma-Aldrich) and FEC (HLA I-restricted peptides from influenza, Epstein–Barr virus and CMV; 1 µg/ml per peptide; Miltenyi Biotec).

The average of two DMSO wells was subtracted from all peptide-stimulated wells for a given PBMC sample and any response that was lower in magnitude than 2 s.d. of these sample specific DMSO control wells was not considered to be a peptide-specific response (given value 0 or 1 on a log scale). Results were expressed as SFCs per $10^6$ PBMC after background subtraction. Results were excluded where positive control wells (ConA and/or FEC) were negative. A responding sample was classed as any sample positive for any of NSP7, 12 or 13.

## Elastic net classification of infection outcomes

Elastic net models were constructed using the eNetXplorer R package v1.1.3[41] to classify uninfected ($n = 16$) versus infected group ($n = 18$) using day −1 nasal soluble mediators (cytokines, chemokines) as predictors. This framework enables the identification of stable models and predictors that are consistently selected across multiple cross-validation (CV) runs. Raw MSD data were first log-normalized to stabilise variance, followed by z-score scaling to standardise each feature to have mean 0 and standard deviation 1. Models were then fitted under the "binomial" family (logistic regression with elastic-net regularisation), with hyperparameters ($\alpha$ and $\lambda$) optimized over 100 repeated 10-fold CV runs. Model performance was evaluated by classification accuracy, and feature importance by the frequency that predictors were selected, i.e., with non-zero coefficients, across model runs. Null distributions of model performance, feature selection frequency, and feature coefficient were generated from 100 runs of random permutation of response labels.

Significance of model performance and feature frequency and coefficient was assessed relative to the null models.

## Conditional independence network analysis

A conditional independence network of baseline immune factors (soluble mediators, immune cell frequencies, RTC-specific T-cell IL-2 response) was inferred to capture direct (i.e., dependencies that remain after statistically conditioning on all other variables), rather than indirect associations. The network was estimated using the Meinshausen-Bühlmann neighborhood selection method[42], in which each variable was fitted on all others using an L1-penalized (Lasso) regression model, with non-zero coefficients indicating conditional dependencies (edges) between variables. Pairwise rank-based associations among baseline features were first summarised in a Spearman correlation matrix which provided sufficient statistics for the regression procedure, and this correlation matrix was then supplied as input to the R huge package v1.3.5[43] (method = "mb") to estimate the network. The penalty hyperparameter $\lambda$ was selected within the range 0-1 at the inflection point of the sparsity-$\lambda$ curve, where network sparsity began to stabilise. For visualization, CD1c+ DC and CD1c- DC cell frequencies, and CCL13 were selected as central nodes, and their first-order neighbors were retained. Scripts are available here https://github.com/ah2426/covhic001.

## Data analysis and statistics

Flow cytometry data were analysed using FlowJo version 10.10.0. Dimension reduction and visualisation was performed by t-distributed stochastic neighbour embedding (t-SNE) and clustering by FlowSOM[44] of downsampled (4000 NK cells per participant) baseline (day -1) concatenated FCS files that were first processed using FlowAI. T-SNE was run with an iteration number of 1000 and perplexity of 30, groups were then pulled out by keyword and sample ID. Statistical analysis was performed using GraphPad Prism version 9.2 and R version 4.2.1 or 4.2.3. Loess plots were generated in R using packages *ggplot, ggplot2, tidyr, ggraph, and ggpubr* with the 'loess' method using the default span=0.75. Comparisons between infected and uninfected groups utilised Mann-Whitney unpaired U tests or ANOVA mixed-effects models with correction and multiple comparisons test indicated in the figure legends. Correlation analyses used Spearman. A p value of less than 0.05 was used to indicate significance.

## Reporting summary

Further information on research design is available in the Nature Portfolio Reporting Summary linked to this article.

## Data availability

Raw sequencing data reused in this paper are accessible through the COVID-19 Cell Atlas web portal https://covid19cellatlas.org and are deposited under controlled access at the European Genome-Phenome Archive under accession number EGAD00001012227. All other data are available in the article and its Supplementary files or from the corresponding author upon request. Source data are provided with this paper.

## Code availability

Analysis code and data for the elastic-net and conditional independence network analyses are available at GitHub (https://github.com/ah2426/covhic001).

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

## Acknowledgements

We thank the study participants for their time and commitment. C.C. and P.O. are supported by the NIHR Imperial Biomedical Research Centre (BRC) award to Imperial College Healthcare NHS Trust and Imperial College. Infrastructure support was provided by the NIHR Imperial BRC and the NIHR Imperial Clinical Research Facility. We thank the following

organisations for their invaluable contributions to development and implementation of the SARS-CoV-2 human challenge project: the Royal Free London NHS Foundation Trust, Human Infection Challenge Network for Vaccine Development (HIC-Vac), NIHR Clinical Research Network staff at the Royal Bolton Hospital, and ISARIC4C Investigators (https://isaric4c.net/about/authors/) for providing the clinical material for challenge virus production. We acknowledge the Laboratory of Medical Sciences (LMS) NIHR flow facility. The views expressed are those of the authors and not necessarily those of the NHS, the NIHR, DHSC or BEIS. The authors gratefully acknowledge support from the UK Vaccine Taskforce of the Department of Business, Energy and Industrial Strategy of Her Majesty's Government (BEIS) and the Wellcome Trust (grant no. 224530/Z/21/Z) and Kwok Foundation. ISARIC4C is funded by the National Institute for Health Research (NIHR; award CO-CIN-01), the Medical Research Council (MRC; grant MC_PC_19059), the NIHR Health Protection Research Unit in Emerging and Zoonotic Infections at University of Liverpool in partnership with Public Health England (PHE), in collaboration with Liverpool School of Tropical Medicine and the University of Oxford (NIHR award 200907), Liverpool Experimental Cancer Medicine Centre provided infrastructure support for this research (grant reference C18616/A25153) and NIHR Health Protection Research Unit in Respiratory Infections (NIHR award 200927). H.R.W. and L.P. acknowledge funding from HIC-Vac. P.O. is supported by an NIHR Senior Investigator Award (NIHR201385) and UKRI MRC CIC Award (MR/V028448/1). L.S. is supported by a Pears Foundation and Rosetrees trust Advancement Fellowship. M.Z.N. acknowledges funding from an MRC Clinician Scientist Fellowship (MR/W00111X/1) and Action Medical Research (GN2911). K.B.W. acknowledges funding from University College London, Birkbeck MRC Doctoral Training Programme and the TrailMap-One Heralth MRC grant (MR/Y03368X/1). L.M.D. is supported by the European Union's Horizon 2020 research and innovation programme under the Marie Skłodowska-Curie grant agreement no. 955321. M.N. acknowledges support from Wellcome Trust (207511/Z/17/Z) and NIHR Biomedical Research Center funding to University College London Hospitals NHS Trust. J.S.T. acknowledges funding from a Chan Zuckerberg Biohub Investigator Award, the NIH (R01AI170116, U01AI153700, U01AI165745, and U19AI145825), and CEPI (Mucosal Immunity in Human Coronavirus Challenge). The funders had no role in the conceptualisation, design, data collection, analysis, decision to publish, or preparation of the manuscript. L.K. is supported by a Royal Society Newton International Fellowship (grant no. NIF-R1-232597).

## Author contributions

H.R.W. and R.S.T. processed samples, acquired, analysed and interpretated data and wrote the manuscript. R.G.H.L., L.K., K.B.W., L.M.D. processed samples, acquired, analysed and interpreted data. J.K.S., L.P., R.M., S.A., A.M.C., J.X., N.L. processed samples and acquired data. B.K., M.K., A.M., and A.C. oversaw sample collection and provided clinical data. L.S. acquired and interpreted data. A.H. and J.S.T. analysed and interpreted data. M.K.M., M.N., M.Z.N., S.A.T., P.J.M.O. interpreted data. C.C. conceived the study, set up the clinical study, directed the study, interpreted data and wrote the manuscript. All authors reviewed and edited the manuscript.

## Competing interests

A.M. and A.C. hold shares in hVIVO Services Ltd. In the past 3 years, S.A.T. has received remuneration for scientific advisory board membership from Sanofi, GlaxoSmithKline, Foresite Labs and Qiagen. S.A.T. is a co-founder and holds equity in Transition Bio and Ensocell. From 8 January 2024, S.A.T. is a part-time employee of GlaxoSmithKline. P.O. declares remuneration for scientific advisory boards from Sanofi, AstraZeneca, GlaxoSmithKline, Pfizer, Seqirus, Moderna, BioNet Asia, Shionogi and IntegerBio. J.S.T. serves on the scientific advisory boards of CytoReason, ImmunoScape, and Snow Medical. All other authors declare no competing interests.

## Additional information

[1]Department of Infectious Disease, Imperial College London, London, UK. [2]National Heart and Lung Institute, Imperial College London, London, UK. [3]Wellcome Sanger Institute, Wellcome Genome Campus, Cambridge, UK. [4]The Netherlands Cancer Institute, Amsterdam, the Netherlands. [5]Cambridge Stem Cell Institute, University of Cambridge, Cambridge, UK. [6]UCL Respiratory, Division of Medicine, University College London, London, UK. [7]Center for Systems and Engineering Immunology (CSEI) and Department of Immunobiology, Yale University, New Haven, CT, USA. [8]Program in Computational Biology and Biomedical Informatics, Yale University, New Haven, CT, USA. [9]Department of Infectious Diseases, University College London Hospital, London, UK. [10]hVIVO Services Ltd., London, UK. [11]Division of Infection and Immunity, University College London, London, UK. [12]Department of Biomedical Engineering, Yale University, New Haven, CT, USA. ✉e-mail: c.chiu@imperial.ac.uk

