## [Transparent Peer Review file · Nature Communications]

Pre-existing and early cellular immune factors correlate with functionally complete protection against primary controlled human SARS-CoV-2 infection

Corresponding Author: Professor Christopher Chiu

Version 0:

Reviewer comments:

Reviewer #1

(Remarks to the Author)

In this study, H. R. Wagstaffe and colleagues investigated immunological correlates of protection (innate and adaptive (T cells and antibodies) in a controlled human infection model of SARS-COV-2 in serologically naïve individuals. This work follows 2 earlier publications where immunological responses were described in the same cohort (Wagstaffe, *Sci Immunology*, 2024 and Lindeboom, *Nature*, 2024). In this new work cross-reactive pre-existing immunity is assessed and some additional detailed analysis into specific innate populations is performed. Strengths are the unique set of patient materials and the detailed analysis of systemic and mucosal immune responses. Some limitations to this work exist and are noted below.

Major

Overall the manuscript seems to lack a common thread. Several immune components are mentioned that sometimes link to protection, but overall understanding of what is causing protection is now not clear after having read this paper and the previous 2 papers. Integrative machine learning could be performed to find whether certain features correlated with each other and/or outcome? Several tools that work with relatively small sample sizes exist and could be used to understand the relative contribution of the different innate and adaptive cell populations that were profiled. Moreover, the novelty related to the previous 2 papers is not immediately clear, perhaps this can be explained better in the discussion. While the other papers are referred to at various points in the manuscript, it would help to have this contextualized clearly.

The authors describe 3 groups of individuals in their cohort: infected, transient infected and abortive (fully protected). For analyses "transient" infected individuals are incorporated sometimes into the fully-protected, while sometimes they are shown as a separate group. This group should be consistently separated out as there are many different analyses performed and now the authors seem to cherry pick which comparison fits the narrative best for a particular figure. Due to the low numbers in the study groups, merging this transient with abortive group might lead to skewing of the results. Please see below specific comments related to the separation of these groups:

1) Fig. 4. The NK cells should be split by the 3 groups, also related to the earlier paper that is cited where this cell population behaves differently between groups. C) Comparing marker expression within clusters between groups is doubtful as the clusters are defined using these markers so any cells with different expression would fall into different clusters. It is also hard to interpret the line plots here as the number of cells differ per donor and can be driven by outliers, so not the most robust way of showing phenotypic changes

2) Lines 164-169; statistics should be performed on uninfected (without transient) vs infected as well. Four out of 5 transient infected individuals have low CCL13 levels (below median), similar to the infected group. Also for CCL22, all 5 transiently infected fall below the median, skewing the data. It is also not clear why the authors specifically looked into these 2 chemokines, while IL-18 was also significantly different (without adjustments).

3) Lines 145-onwards: Baseline antibodies; protected is now considering both transient and uninfected, however, looking at the lining fluid data; it seems that the fully protected abortive group has higher IgA and the transient infected individuals are at the same level as infected, both for HKU1, SARS1 and SARS2. The comparison of fully protected versus infected should be done to show that the transient group is not responsible for the absence of significance.

Related to the comparisons to the different groups, please see specific comments below where only “observations” are mentioned within specific groups, without comparing to another group or a specific group was left out to compare it with.

- 1) Lines 259-263; The authors have only looked at this in the infected group, it could be speculated that an even higher increase of this population would be observed in the protected group, hence their reported association with protection. The comparison here with the protected group needs to be made to strengthen the data in this paragraph. Also the figure legends of the extended data does not match the figure. Please add a legend there that the red circles are infected individuals.
- 2) Lines 185-188. Can the authors state whether there are differences between infected and uninfected for these chemokine receptors? Are they correlated with ligands and/or recruitment to the mucosa?
- 3) Lines 148; only correlations with nasal antibodies was performed. Please add the same for plasma-derived.

The data presented about the differences of NK-cell subsets need some revision to allow proper interpretation of the results. Please see below specific comments related to this:

- 1)Line 234-235: It would be helpful to point to Extended data 4A here to show where to find the nasal NK cell data. But looking at it they seem to express NKG2C, so this does not match the statement from the text.
- 2)Is 5C statistically significant? Statistical tests should be done and reported.
- 3)The frequency of cluster 1 NK cells is not shown as boxplot with individuals, just with density plot. Are these diffs significant?
- 4)Line 231 & 232-235: this is not true, since cluster 2 from FLOWsom is a very large cluster and does not only contain the highlighted cluster in a.
- 5)Any granzymes or cytokines being produced based on the single cell data in the NK? Given the speculation on cytotoxicity and functionality based on their phenotype

Lines 171-174: fig 3c does not match with the statistical test performed. Moreover, the conclusion in results and discussion that baseline levels of CCL13 and CCL22 associate with the recruitment of DCs that then are major producers of the chemokines leading to mutual reinforcement does not seem to match the longitudinal protein data as CCL13/CCL22 are not increased over time in vivo (despite the induction at an RNA level in DCs).

Fig 3g. Additional data should be supplemented to make statements about the expression of CCL22 in total of 14 cells.

Lines 275-279 and figure 5a; it is unclear how these data are different from what was published in the previous Nature paper?

Were the 6 protected individuals with relatively high levels of nucleocapsid IgG pre-exposed to SARS-CoV-2 or is this cross-protective? What are the antibody titers against other seasonal coV N proteins? Measuring this could indicate whether this is indeed cross-protection.

Is the correlation between IgM spike in 2c corrected for multiple testing (as 27 tests are done?)

Figure 2D, is there a correlation of the IgM with duration or load, or only with the first positive PCR that then correlates with the number of positive days?

Minor comments

Abstract; “infiltration” of tissue resident T cells. This has not been studied. It could be proliferating Trms, re-circulating Trms, Tem infiltration . Suggestion to put “increase in numbers of Trms’ instead
Extended data 3 seems to gate both live and dead cells.

Line 182; Suggest to rephrase, it is now confusing whether “lacking” just refers to day 1 or lacking at all timepoints in the infected

Lines 204; “enhanced antigen presentation”, since the authors have single-cell data, they should be able to look into HLA-class-I upregulation or pathways associated with this. The authors have already found DQ2 to be associated in their Nature paper, related to this data, although class-II, there might be valuable insights to be further explored here.

Lines-280-283: please add baseline “peripheral” to the text earlier (now at line 288) Without this, the authors imply here that they measured these responses derived from nasal material, which is not the case Please add the number of samples for each group here tested as well (line293).

Lines 206: “In particular, while not powered to test differences between these subgroups, there was no evidence that CCL13, CCL22 or DC frequencies were elevated in transiently infected individuals” What is the study powered for? Is it powered to detect such frequencies even between the combined groups?

Fig 5a. can the authors please, similarly as before, add the number of cells from each of the subsets that were analysed?

Lines 262-270: there is a shift in figure calling, figure 4h, should be g. 4i should be h, 4j should be i.

Reviewer #2

(Remarks to the Author)

This manuscript by Wagstaffe and colleagues expands on important observations made in participants from a controlled human SARS-CoV-2 challenge study that were previously reported (primarily PMIDs: 38335268 and 38898278) by the same group. In contrast to earlier work, this manuscript focuses on the immunologic characteristics of the uninfected individuals from the study to identify features of subjects with no known prior infection with SARS-CoV-2 that resisted infection following challenge. The investigators inoculated 34 seronegative young adults with a pre-Alpha variant SARS-

CoV-2 stock and identified three distinct infection outcomes: sustained infection (N=18), transient infection (N=5), and abortive infection (N=11). Key findings reported in this manuscript include the observation that elevated nasal CCL13 levels at baseline were associated with protection from infection, suggesting innate chemokine-mediated recruitment is a critical first-line defense. In addition, individuals who resisted infection showed enrichment of less differentiated NK cells (CD56dimCD57-NKG2C-) and higher frequencies of T cells specific for viral non-structural proteins. The methods are robust, and the manuscript is well-written.

The following suggestions will help enhance the manuscript prior to publication:

- 1) Extended Data Figure 4d and 4e – the legend for figure 4d appears to be absent and the legend that states it is for 4d appears to coincide with panel 4e. Please adjust.
- 2) Discussion lines 347-349, Figure 2a, Extended Data Figure 2 – the substantially elevated nucleocapsid plasma IgG level in these six individuals that are all in the abortive infection group in the absence of spike-specific IgG responses may not be a clear indicator that they were previously infected, as the authors note. However, “unlikely that these higher levels were due to a previously undetected SARS-CoV-2 infection” is strongly worded. It is possible that these subjects either 1) had a very mild previous infection relatively remotely to study entry and therefore spike IgG responses had waned; or 2) these individuals may have experienced previous transient infection following an earlier exposure event resulting in partial adaptive immune response development. The discussion should acknowledge that although these six participants appear very similar to the other abortive infection participants in other immunologic measures, they may represent a unique subset of the abortive infection cohort and not state so definitively that these 6 were infection naïve at study entry.
- 3) The discussion should acknowledge limitations of the study cohort with regards to the relatively young age of the cohort. The discussion should also acknowledge the limitation of the very early virus isolate that was studied that may or may not have the same immune evasion properties as currently circulating viral isolates in the context of the immune responses that are associated with protection in this study.

Reviewer #3

(Remarks to the Author)

The study aims to understand the immune signatures that enable some humans to resist SARS-CoV-2 infection naturally. Healthy individuals who had not been previously infected with the virus were inoculated with the virus, and nasopharyngeal swabs for cells, as well as PBMCs, were analysed in conjunction with symptom testing and qPCR positivity. Single-cell transcriptomics were performed on the cells to identify DCs/monocytes, NK cells, chemokines, and cross-reactive T cells that could be important for resistance to the virus locally without raising the alarm for a systemic effect. The extent of the scRNA-seq data analysis is great, to identify the different characteristics of the cell clusters and to correlate with the infection outcome, as well as cell analysis with different experiments.

Interesting findings were reported for those resistant individuals at baseline:

1. Increased cDCs or migrated monocytes in the nasopharynx.
2. Increased CCL13 at the site of inoculation.
3. Increased less differentiated NK cells.
4. Presence of cross-reactive T cells against the non-structural proteins of SARS-CoV-2.

The well-controlled and designed study is worthwhile for insights into the immune cells/combination that may ensure protection from the virus at the mucosal site. My major concern is how to tie these factors together.

Major comments:

-the DCs/monocytes, NK cells, CCL13, and cross-reactive memory T cells are the factors. It is entirely possible that tissue-resident DCs could respond to the virus inoculation, release CCL13 to attract monocytes and cross-reactive memory T cells to the act, and these DCs can perform antigen presentation to the T cells; the tissue-resident CD57- NK cells may cross-talk with the resident DCs or monocytes, while they, themselves, could be further releasing cytokines to enhance the Th1 response. There is currently no discussion on their combined roles or proposing a model for their interactions to go against SARS-CoV-2.

-I understand that it is difficult to obtain the human specimens and PBMCs after the study has been conducted. But I do wonder if in vitro experiments could be performed to mix a combination of these cells and test the enhanced effect on killing NSP-expressing target cells. Could they be even more effective for the uninfected individuals at baseline, compared to the infected individuals with the newly raised adaptive immune response?

-I also wonder how these DCs, NK or cross-reactive T cells would compare to vaccinated individuals for protection. Perhaps the authors can discuss on this.

-The antibodies tested are against spike or NP, while the cross-reactive T cells are responsive to NSP7, NSP12, and NSP13. Is it possible that pre-existing antibodies against the NSP peptides (or RTC) could play a role in resisting the infection?

-If no antibodies are involved, and the cDC1/monocytes could be the major antigen presenting cells, would there be

sufficient NSP/RTC proteins for presentation when the infection is controlled in a timely manner without the chance for viral replication (or low undetectable level)? Would it mean that these antigens are even more immunodominant than spike proteins or NP?

Minor comments:

-Line 197: The authors suggest that cDCs could be producing CCL13 once they traffic into the site, could there be CCL13 produced by tissue-resident DCs?

-The role of IL-18, as shown in Extended Data Table 1 should be discussed, as it can enhance NK functions.

-Titles for figure legends for Fig. 5 and Extended Data Fig. 3 had formatting issues where it is bulged by the figures.

-Should Fig. 1b go before 1a?

-Cluster 1 in Fig. 4a and 4c are described in the text. Did I miss cluster 2?

-Extended Data Fig. 3, is a plot missing in panel d first row of FACS plots where the arrow is pointing?

-Line 561, the flurophore for CD3 is missing

Reviewer #4

(Remarks to the Author)

Reviewer #5

(Remarks to the Author)

Version 1:

Reviewer comments:

Reviewer #1

(Remarks to the Author)

All comments have been addressed

Reviewer #2

(Remarks to the Author)

The authors have addressed all concerns raised in the initial round of reviews and the revised manuscript is appropriate for publication.

Reviewer #3

(Remarks to the Author)

The authors have adequately addressed my comments. The revised manuscript is significantly improved with the additional data and the presentation of them.

Reviewer #4

(Remarks to the Author)

Reviewer #5

(Remarks to the Author)

I co-reviewed this manuscript with one of the reviewers who provided the listed reports. This is part of the Nature

Communications initiative to facilitate training in peer review and to provide appropriate recognition for Early Career Researchers who co-review manuscripts.

Please find our point-by-point response to the reviewers' comments below.

Reviewer #1 (Remarks to the Author):

In this study, H. R. Wagstaffe and colleagues investigated immunological correlates of protection (innate and adaptive (T cells and antibodies) in a controlled human infection model of SARS-COV-2 in serologically naïve individuals. This work follows 2 earlier publications where immunological responses were described in the same cohort (Wagstaffe, Sci Immunology, 2024 and Lindeboom, Nature, 2024). In this new work cross-reactive pre-existing immunity is assessed and some additional detailed analysis into specific innate populations is performed. Strengths are the unique set of patient materials and the detailed analysis of systemic and mucosal immune responses. Some limitations to this work exist and are noted below.

We thank the reviewer for their supportive comments and hope our response have addressed these limitations.

Major

1. Overall the manuscript seems to lack a common thread. Several immune components are mentioned that sometimes link to protection, but overall understanding of what is causing protection is now not clear after having read this paper and the previous 2 papers. Integrative machine learning could be performed to find whether certain features correlated with each other and/or outcome? Several tools that work with relatively small sample sizes exist and could be used to understand the relative contribution of the different innate and adaptive cell populations that were profiled.

We thank the reviewer for suggesting this approach and as advised, we have now undertaken multivariate and network analyses using machine learning to find whether features correlated with each other and outcome. First, we employed logistic regression with elastic-net regularisation using the entire baseline soluble mediator dataset to model whether a multivariate classifier was capable of predicting risk of infection. Next, we undertook conditional independence network analysis using all the preceding baseline immune data to assess the inter-relatedness and dependencies of these disparate arms of immunity. In this way, a putative regulatory structure was revealed, including a subnetwork involving CCL13 as a hub connecting pre-existing RTC-specific T cell responses to CCL13 via CD1c⁺ DCs. This is now in a new Figure 6 and text results lines 372-406 methods lines 689-719 We have also added a section to the discussion, incorporated with our response to reviewer 3, proposing a model (discussion lines 537-545). We believe that this now provides a common thread that draws together the interpretation of these data.

1.1. Moreover, the novelty related to the previous 2 papers is not immediately clear, perhaps this can be explained better in the discussion. While the other papers are referred to at various points in the manuscript, it would help to have this contextualized clearly.

At the reviewer's recommendation, we have now revised to make more explicit how this paper differs from previous analyses of the same cohort (mainly introduction lines 100-107). Specifically, the current manuscript focuses on pre-inoculation differences between infected and uninfected participants in order to identify potential correlates and mechanisms of protection, with analysis of post-inoculation responses only where they arise directly from these baseline differences. Previous papers have not presented any baseline immune differences, focusing only on post-inoculation responses and their relationship with clinical outcomes.

2. The authors describe 3 groups of individuals in their cohort: infected, transient infected and abortive (fully protected). For analyses “transient” infected individuals are incorporated sometimes into the fully-protected, while sometimes they are shown as a separate group. This group should be consistently separated out as there are many different analyses performed and now the authors seem to cherry pick which comparison fits the narrative best for a particular figure. Due to the low numbers in the study groups, merging this transient with abortive group might lead to skewing of the results.

We thank the reviewer for highlighting the importance of careful grouping for analysis of these clinical outcomes. In our earlier manuscript, we showed that transient and abortive infection groups were associated with subtly different immune responses in the upper respiratory tract detectable only at the single-cell RNA level (Lindeboom et al Nature, 2024). However, neither group met the protocol-defined criteria for infection, developed symptoms or seroconverted. In clinical practise, these two outcomes would therefore be virtually indistinguishable. We therefore decided to focus primarily on comparing sustained infection with the combined group of “functionally complete protection”, which we believe to be the most clinically relevant.

To clarify this and ensure comparisons are consistent throughout, we now state the rationale explicitly (lines 133-137 and 423-427) and have presented only 2-group analyses in the main figures (Figures 2-6), with 3-group analyses (which ask a somewhat different question with reduced power due to smaller group sizes) restricted to the extended data (Extended Data Figures 1b-d, 3f-h, 4e, 5a and 6c, 7b, c). To maintain transparency, we have used symbols to distinguish the transient and abortive infections within the uninfected group in the antibody, chemokine and now NK cell and T cell figures (newly added to Figures 4e and 5d).

Where we have made 3-group comparisons in the extended data, we saw no differences between the transient and abortive or infected groups, except in the antibody levels as noted below. Importantly, key differences in CCL13, DC and NK cells that we describe remain statistically significant in comparing sustained infection with the uninfected (abortive) group when the transient infections are removed, supporting the robustness of these findings. Thus, we have made consistent the groupings throughout the manuscript, avoiding potential cherry-picking, and shown that the main findings are not skewed by merging transient with abortive infections.

2.1. Fig. 4. The NK cells should be split by the 3 groups, also related to the earlier paper that is cited where this cell population behaves differently between groups.

Due to the limited number of cells remaining available for this analysis, only 4 transiently-infected participants could be included in the NK cell data. However, longitudinal plots of NK cell subset and proliferation data separated into three groups are now added to Extended Data Figure 4e & 5a. There were no significant differences between the transient and abortive or infected groups when a 3-group comparison was undertaken, reiterating that the subtly different responses shown in the transient group were only detectable in the local mucosa at RNA level.

2.2. Comparing marker expression within clusters between groups is doubtful as the clusters are defined using these markers so any cells with different expression would fall into different clusters. It is also hard to interpret the line plots here as the number of cells differ per donor and can be driven by outliers, so not the most robust way of showing phenotypic changes

We thank the reviewer for these comments, which have provided direction on how to improve this section. To generate the t-SNE plot, down-sampling and concatenation of individual data files (as

explained in Materials and Methods lines 724-725) was used to include the same number of NK cells from each participant in the map. t-SNE analysis then identified clusters of events based on their relatedness by multiple phenotypic variables. Thus, interpretation of the line plots in Figure 4c should not have been affected by differing numbers of cells per donor and represents aggregated results from multiple donors, thus mitigating for outliers. While clusters are identified for their similarity of marker expression by this unsupervised analysis (with Cluster 1 in Figure 4 overall high for CD57, CD16 and NKG2C), the line plots reveal further within-cluster heterogeneity of expression that distinguished infected and uninfected participants. We believe this to be a meaningful finding.

To improve the clarity of message in this section, the t-SNE plots have been reanalysed and further data has been added to Figure 4. We now show the frequency, count and MFI of FlowSOM cluster 1 in addition to the line plots and the frequency of all other clusters in Extended Data Figure 4. We apologise for the missing axis labels on the line plots of Figure 4c; these have now added, with the figure thus showing both the differential expression level of NKG2C and FC ϵ R1 γ , and the count of high- and low-expressing cells in the cluster. We have edited the text to explain this (lines 260-270).

2.3. Lines 164-169; statistics should be performed on uninfected (without transient) vs infected as well. Four out of 5 transient infected individuals have low CCL13 levels (below median), similar to the infected group. Also for CCL22, all 5 transiently infected fall below the median, skewing the data. It is also not clear why the authors specifically looked into these 2 chemokines, while IL-18 was also significantly different (without adjustments).

Thank you for this suggestion. These statistics had been performed on the full time-course and described on line 223. The data are shown in Extended Data Figure 3f and differences were non-significant. Additional statistics have now been performed on the baseline timepoint alone as suggested, and the significant difference in CCL13 level was upheld in the abortive vs the infected groups. This has been added to Extended Data Figure 3g and line 225. This is in line with our statement 'there was no evidence that CCL13, CCL22 or DC frequencies were elevated in transiently infected individuals' (line 226). We also note that the transient group is more in line with the infected than the abortive group for these measures (line 227).

Of the soluble mediators evaluated, indeed both CCL13 and IL-18 showed significant differences at day 0 however, the difference in IL-18 only just reached statistical significance ($p=0.0424$) without correction for multiple comparisons, with higher levels in the infected participants only at day 0 and not day -1 as well, suggesting variability of IL-18 at these baseline timepoints. We therefore focused on CCL13 and CCL22 due to association with protection and similar pattern of expression, with consistently higher levels in the nasal lining fluid of uninfected individuals up to 5 days post-inoculation which was distinct from IL-18. Furthermore, IL-18 did not form part of the co-regulated cluster in the later network analysis. We have now explained the rationale more clearly in the text (lines 232-240) and included the Loess plot of IL-18 expression in Extended Data Figure 3i showing the divergent expression pattern and Extended Data Table 3, which shows the mediator p -values at the additional baseline timepoint of day -1.

2.4. Lines 145-onwards: Baseline antibodies; protected is now considering both transient and uninfected, however, looking at the lining fluid data; it seems that the fully protected abortive group has higher IgA and the transient infected individuals are at the same level as infected, both for HKU1, SARS1 and SARS2. The comparison of fully protected versus infected should be done to show that the transient group is not responsible for the absence of significance.

We thank the reviewer for highlighting these differences between groups. Analysis of the baseline antibody data by 3 groups has now been carried out. With correction for multiple antigen and isotype comparisons, there was indeed a significantly higher anti-SARS-CoV-2 S IgA level in the abortive vs the transient group in nasal lining fluid, with no difference between the abortive and infected groups. However, there remains no significant difference between either the abortive nor transient groups when compared with infected individuals. This analysis has been added to Extended Data Figure 2b and results lines 153-161 and discussion lines 431-446.

3.1. Lines 259-263; The authors have only looked at this in the infected group, it could be speculated that an even higher increase of this population would be observed in the protected group, hence their reported association with protection. The comparison here with the protected group needs to be made to strengthen the data in this paragraph.

In the previously published single cell RNA transcriptomic data, we showed evidence of proliferation (NK cycling) in the nasal cells of the infected group and at day 1 in the transient group but not the abortive group (Lindeboom et al Nature, 2024). Here, we now include plots showing the expression of the proliferation marker Ki67 in all 3 groups in Extended Data Figure 5a. In this analysis, no peripheral NK cell proliferation was observed by flow cytometry in either the transient or abortive groups, potentially due to the lack of day 1 PBMC sampling for this analysis, the lower sensitivity of this method or because this response was limited to the local mucosa. This has been added to lines 301-302.

3.1.1. Also the figure legends of the extended data does not match the figure. Please add a legend there that the red circles are infected individuals.

Many thanks for pointing this out. The figure legend has been edited to match the figure and a legend showing red circles as infected has been added into the Extended Data Figure.

3.2 . Lines 185-188. Can the authors state whether there are differences between infected and uninfected for these chemokine receptors? Are they correlated with ligands and/or recruitment to the mucosa?

Thank you for suggesting we look deeper into these data. To answer this question, we interrogated the scRNAseq data for differential baseline expression of chemokine receptor genes by myeloid cells between infected and uninfected groups. This showed a significant (Mann-Whitney $p=0.042$) difference in CCR1 expression, which was higher in infected individuals. While CCR1 does recognise CCL13, it also has promiscuous binding to multiple other chemokines and it is therefore difficult to draw any conclusions about causal relationships during this later part of the response to infection. We attach the figure showing this analysis as Figure 1 for reviewers.

To address the second point, we tested the correlation between overall CCL13 expression in nasal myeloid cells and chemokine receptor gene expression in circulating or nasal myeloid cells. There was no positive correlation between nasal CCL13 and circulating CCR1 expression in the uninfected group at any time. There were non-significant positive correlations between nasal CCL13/22 and nasal CCR1 as well as circulating CCR2/4. While these may be suggestive of chemotactic signalling to the nasal mucosa, we believe that interpretation of these limited data would be overly speculative, given the multiple ligands that each of these chemokine receptors recognise. We have included this analysis as Figure 2 for reviewers but with the relevance of both analyses so unclear, we have elected not to include these data in the revised paper at present.

3.3. Lines 148; only correlations with nasal antibodies was performed. Please add the same for plasma-derived.

Thank you for this suggestion. The correlations have been performed and there were no significant correlations between the time to qPCR positivity and baseline plasma antibody levels. The plasma correlation plot has been added to Figure 2c, line 167.

4.1. Line 234-235: It would be helpful to point to Extended data 4A here to show where to find the nasal NK cell data. But looking at it they seem to express NKG2C, so this does not match the statement from the text.

Thank you for this suggestion. We have now added a reference to Extended Data 4a here and have changed it to indicate a 'similar' phenotype to the nasal NK cells, which was NKG2C^{low}.

4.2. Is 5C statistically significant? Statistical tests should be done and reported.

We have understood this to mean Figure 4c (as under a subheading of NK cells). The MFI of cluster 1 from the new analysis has now been plotted and statistics performed. We have kept the line plots showing the average count vs MFI of NKG2C and FC ϵ R1 γ as these plots show the greater NKG2C positive cell count and MFI combined and show the population of FC ϵ R1 γ negative cells.

If this was relating to Figure 5c, Mann-Whitney U test was carried out and stated in the figure legend; we have now added lines to the plots indicating the non-significant comparisons.

4.3. The frequency of cluster 1 NK cells is not shown as boxplot with individuals, just with density plot. Are these diffs significant?

This figure has now been revised in response to comment 2.2 above. In the new version, the counts (Figure 4b) and frequencies (Extended Data Figure 4c) are shown with all individuals and statistics have been performed.

4.4. Line 231 & 232-235: this is not true, since cluster 2 from FLOWsom is a very large cluster and does not only contain the highlighted cluster in a.

Thank you for highlighting this. These data have been reanalysed and rephrased in response to comment 2.2 above.

4.5. Any granzymes or cytokines being produced based on the single cell data in the NK? Given the speculation on cytotoxicity and functionality based on their phenotype

We thank the reviewer for the opportunity to develop this section further. We have now interrogated the single cell data to look into expression of cytokine and cytotoxicity modules of NK cells in the nose. The plots show an early activated and cytokine signature that predominates over cytotoxicity in the abortive infection group, whereas cytotoxicity is more dominant in the sustained and transient infection groups. This figure has been added as Extended Data Figure 4g (lines 294-298) to support the conclusion that enrichment of less differentiated, cytokine responsive/producing NK cells at baseline and early after inoculation, was associated with resisting infection.

5.1. Lines 171-174: fig 3c does not match with the statistical test performed.

Thank you for this comment, a table showing the adjusted multiple Mann-Whitney comparisons of CCL13 level between infected and uninfected groups by day post-inoculation has been added as Extended Data Table 2 and added to the text line 191.

5.2 Moreover, the conclusion in results and discussion that baseline levels of CCL13 and CCL22 associate with the recruitment of DCs that then are major producers of the chemokines leading to mutual reinforcement does not seem to match the longitudinal protein data as CCL13/CCL22 are not increased over time in vivo (despite the induction at an RNA level in DCs).

We agree that this is an interpretation of the data and therefore a hypothesis rather than conclusion. There are a number of potential explanations for why CCL13 and CCL22 protein levels do not increase over time despite the transcriptional response. The half-life of chemokines is generally short, of the order of minutes to hours, although this is not been clearly shown for CCL13 and CCL22. Rapid turnover may be mediated by proteolytic cleavage, post-translational modifications, or receptor-mediated internalisation and degradation. Testing of these is beyond the capabilities of this study but we have revised the text (results line 229, discussion lines 537-545) clearly stating that this is a hypothetical model.

6. Fig 3g. Additional data should be supplemented to make statements about the expression of CCL22 in total of 14 cells.

Thank you for suggesting we revise this section. To avoid over-interpretation, we have removed this row in the figure and replaced it with DCs expressing CD69 and CD103, therefore displaying a tissue resident phenotype. With 101 and 61 such cells in the infected and uninfected respectively, this analysis is more robust. The text has been altered accordingly (lines 200-205).

7. Lines 275-279 and figure 5a; it is unclear how these data are different from what was published in the previous Nature paper?

This figure differs from the previous Nature paper in 2 ways: this is the only presentation of these data as 2 groups (infected and uninfected only) as opposed to the 3 groups presented in the Nature paper and in the extended data figure of this paper. In addition, the annotation granularity is one level more detailed than that previously presented, making this figure a more detailed analysis of the T cell subsets shown.

8. Were the 6 protected individuals with relatively high levels of nucleocapsid IgG pre-exposed to SARS-CoV-2 or is this cross-protective? What are the antibody titers against other seasonal coV N proteins? Measuring this could indicate whether this is indeed cross-protection.

Our data suggest that this is most likely cross-reactive anti-N IgG from previous seasonal coronavirus exposure rather than unrecognised pre-exposure to SARS-CoV-2. Anti-N IgG levels in the plasma of participants following sustained infection reached levels of over 45,000AU/ml post-inoculation (see Figure 3 for reviewers), whereas the median concentration in the abortive group at baseline was 559.9AU/ml when measured using the same MSD platform. These 6 participants did not have higher levels of anti-S antibody compared with the other uninfected participants and it has been shown that anti-N antibody levels wane more quickly than anti-S antibody levels (Grandjean et al. Clin Infect Dis, 2022), so it is reasonable to have expected a

heightened level of spike antibody in these individuals if the pre-existing N antibody was due to a prior SARS-CoV-2 infection. These participants also did not display T cell responses measured by ELISpot in response to S or N at baseline. The MSD platform does not allow comparable analysis of seasonal CoV N protein responses but, together, the above suggests previous infection is unlikely. Nevertheless, previous asymptomatic SARS-CoV-2 exposure without seroconversion is still a possibility that cannot be absolutely ruled out in this cohort. It is indeed possible individuals might have had a transient infection previously resulting in some immune responses but without seroconversion, much as we have seen in this study. We have edited the results and discussion section to discuss this possibility (results lines 153-161 and discussion lines 431-446)

9. Is the correlation between IgM spike in 2c corrected for multiple testing (as 27 tests are done?)

Yes, it is corrected (FDR); this is explained in the figure legend.

10. Figure 2D, is there a correlation of the IgM with duration or load, or only with the first positive PCR that then correlates with the number of positive days?

There was no correlation between baseline antibody level and viral load or duration qPCR positivity.

Minor comments

1. Abstract; “infiltration” of tissue resident T cells. This has not been studied. It could be proliferating Trms, re-circulating Trms, Tem infiltration . Suggestion to put” increase in numbers of Trms’ instead

Many thanks for this suggestion, this has been edited to ‘increased numbers’ (line 39).

2. Extended data 3 seems to gate both live and dead cells.

Thank you for suggesting we check this figure. Extended Data Figure 3d is RBC lysed whole blood, therefore this plot includes granulocytes, monocytes and lymphocytes which are spread out along the FCS axis. In Extended Data Figure 4b, we use the same viability dye on isolated PBMC samples, and the live population is similarly up to just over 10^3 on the y axis (Zombie UV). We have checked the raw data and are therefore confident of the gating of live cells in Extended Data Figure 3d.

3. Line 182; Suggest to rephrase, it is now confusing whether “lacking” just refers to day 1 or lacking at all timepoints in the infected

Thank you, this has been rephrased. This now includes that CCL13 was lacking at day 1 in the infected group, but was measured at day 7 and 10 in the infected group (lines 202-204).

4. Lines 204; “enhanced antigen presentation”, since the authors have single-cell data, they should be able to look into HLA-class-I upregulation or pathways associated with this. The authors have already found DQ2 to be associated in their Nature paper, related to this data, although class-II, there might be valuable insights to be further explored here.

There was no detectable difference in HLA-class I gene expression at single cell level between the 3 infection outcome groups, across all timepoints. The main difference was therefore in DC frequencies. Please see Figure 4 for reviewers.

5. Lines 280-283: please add baseline “peripheral” to the text earlier (now at line 288) Without this, the authors imply here that they measured these responses derived from nasal material, which is not the case. Please add the number of samples for each group here tested as well (line 293).

Thank you, this has been added. The number of samples for each group tested have also been added to the text as well as already being in the figure legend (line 332).

6. Lines 206: “In particular, while not powered to test differences between these subgroups, there was no evidence that CCL13, CCL22 or DC frequencies were elevated in transiently infected individuals” What is the study powered for? Is it powered to detect such frequencies even between the combined groups?

This study was a first-in-human, characterisation study that aimed to develop a safe and reproducible infection challenge model. The primary objectives were safety and infection rate. Therefore, the clinical study was not *a priori* powered to detect differences in any immunological measurements, with these being exploratory endpoints. Nevertheless, experimental medicine studies such as this have been used extensively to test for differences in immune response following infection and been able to identify statistically significant differences. Using the combined groups, significant differences in CCL13, DC and NK cell subsets were detected, with further support for their relevance provided by the additional machine learning analysis suggested by the reviewer.

7. Fig 5a. can the authors please, similarly as before, add the number of cells from each of the subsets that were analysed?

Thank you, the number of cells has been added to Fig. 5a and Extended Data Fig 6c.

8. Lines 262-270: there is a shift in figure calling, figure 4h, should be g. 4i should be h, 4j should be i.

Thank you, this figure legend has been corrected.

Reviewer #2 (Remarks to the Author):

This manuscript by Wagstaffe and colleagues expands on important observations made in participants from a controlled human SARS-CoV-2 challenge study that were previously reported (primarily PMIDs: 38335268 and 38898278) by the same group. In contrast to earlier work, this manuscript focuses on the immunologic characteristics of the uninfected individuals from the study to identify features of subjects with no known prior infection with SARS-CoV-2 that resisted infection following challenge. The investigators inoculated 34 seronegative young adults with a pre-Alpha variant SARS-CoV-2 stock and identified three distinct infection outcomes: sustained infection (N=18), transient infection (N=5), and abortive infection (N=11). Key findings reported in this manuscript include the observation that elevated nasal CCL13 levels at baseline were associated with protection from infection, suggesting innate chemokine-mediated recruitment is a critical first-line defence. In addition, individuals who resisted infection showed enrichment of less differentiated NK cells (CD56dimCD57-NKG2C-) and higher frequencies of T cells specific for viral non-structural proteins. The methods are robust, and the manuscript is well-written.

We thank the reviewer for their supportive comments.

The following suggestions will help enhance the manuscript prior to publication:

1) Extended Data Figure 4d and 4e – the legend for figure 4d appears to be absent and the legend that states it is for 4d appears to coincide with panel 4e. Please adjust.

Thank you, this has been rectified.

2) Discussion lines 347-349, Figure 2a, Extended Data Figure 2 – the substantially elevated nucleocapsid plasma IgG level in these six individuals that are all in the abortive infection group in the absence of spike-specific IgG responses may not be a clear indicator that they were previously infected, as the authors note. However, “unlikely that these higher levels were due to a previously undetected SARS-CoV-2 infection” is strongly worded. It is possible that these subjects either 1) had a very mild previous infection relatively remotely to study entry and therefore spike IgG responses had waned; or 2) these individuals may have experienced previous transient infection following an earlier exposure event resulting in partial adaptive immune response development. The discussion should acknowledge that although these six participants appear very similar to the other abortive infection participants in other immunologic measures, they may represent a unique subset of the abortive infection cohort and not state so definitively that these 6 were infection naïve at study entry.

We agree that there is a theoretical possibility that some of the participants may have had prior SARS-CoV-2 exposure resulting in no symptoms or detectable spike-specific antibody or T cell responses. The results and discussion have been edited in response to this and a similar comment from reviewer 1. Specifically, the discussion has been toned down to acknowledge that transient SARS-CoV-2 infection without seroconversion is possible (lines 431-446) and that this could be one explanation for relative protection in this subset of “protected” individuals.

3) The discussion should acknowledge limitations of the study cohort with regards to the relatively young age of the cohort. The discussion should also acknowledge the limitation of the very early virus isolate that was studied that may or may not have the same immune

evasion properties as currently circulating viral isolates in the context of the immune responses that are associated with protection in this study.

Many thanks for this suggestion; this has been added to the discussion (lines 547-549).

Reviewer #3 (Remarks to the Author):

The study aims to understand the immune signatures that enable some humans to resist SARS-CoV-2 infection naturally. Healthy individuals who had not been previously infected with the virus were inoculated with the virus, and nasopharyngeal swabs for cells, as well as PBMCs, were analysed in conjunction with symptom testing and qPCR positivity. Single-cell transcriptomics were performed on the cells to identify DCs/monocytes, NK cells, chemokines, and cross-reactive T cells that could be important for resistance to the virus locally without raising the alarm for a systemic effect. The extent of the scRNA-seq data analysis is great, to identify the different characteristics of the cell clusters and to correlate with the infection outcome, as well as cell analysis with different experiments.

Interesting findings were reported for those resistant individuals at baseline:

1. Increased cDCs or migrated monocytes in the nasopharynx.
2. Increased CCL13 at the site of inoculation.
3. Increased less differentiated NK cells.
4. Presence of cross-reactive T cells against the non-structural proteins of SARS-CoV-2.

The well-controlled and designed study is worthwhile for insights into the immune cells/combinations that may ensure protection from the virus at the mucosal site. My major concern is how to tie these factors together.

We thank the reviewer for their supportive comments and hope our responses have addressed this major concern.

Major comments:

1. The DCs/monocytes, NK cells, CCL13, and cross-reactive memory T cells are the factors. It is entirely possible that tissue-resident DCs could respond to the virus inoculation, release CCL13 to attract monocytes and cross-reactive memory T cells to the act, and these DCs can perform antigen presentation to the T cells; the tissue-resident CD57- NK cells may cross-talk with the resident DCs or monocytes, while they, themselves, could be further releasing cytokines to enhance the Th1 response. There is currently no discussion on their combined roles or proposing a model for their interactions to go against SARS-CoV-2.

We thank the reviewer for this feedback. In response to this and reviewer 1's comment, we have now undertaken predictive modelling and network analysis using machine learning to identify protective factors in a multivariate model and then infer their inter-relatedness and dependencies. This is now in a new Figure 6 and text results lines 372-406, methods lines 689-719 and discussion lines 537-545 and indicates the statistical relationships between these highlighted immune factors.

2. I understand that it is difficult to obtain the human specimens and PBMCs after the study has been conducted. But I do wonder if *in vitro* experiments could be performed to mix a combination of these cells and test the enhanced effect on killing NSP-expressing target cells. Could they be even more effective for the uninfected individuals at baseline, compared to the infected individuals with the newly raised adaptive immune response?

We agree that *in vitro* experiments to further investigate the functionality of NSP-specific T cells would be informative. Unfortunately, very few PBMC samples remain that could be used to carry out such additional experiments. We have considered using 6 month or 1 year follow-up samples or asking participants to return for further sampling, but these would not be appropriate due to the

high rates of COVID-19 following the quarantine phase of this study, resulting in a SAR-CoV-2 infection/immunological landscape that has changed dramatically thereafter.

Some of what the reviewer suggests has already been published. The ability of clonally expanded TCRs that recognise NSP12 in SARS-CoV-2 unexposed individuals to kill a K562 cell line expressing NSP12 has been shown (Nesterenko et al Cell Rep 2021), demonstrating natural processing of CD8⁺ T cell epitopes in NSP12, and the ability of pre-existing cross-reactive TCRs to recognise NSP12. As discussed in more detail below, NSP12-specific T cells may be more effective at early recognition and control of viral replication due to their ability to recognise the first viral protein to be made in an infected cell (ORF1ab), in particular if there is a delay to the expression of the sub-genomic RNAs that encode the structural proteins such as S and N. Although a lack of baseline PBMC precludes direct evidence being generated in these specific individuals, there is evidence to suggest that the RTC-specific T cells could have an advantage over *de novo* T cells generated during a detectable infection. We have added this to the discussion (line 492).

3. I also wonder how these DCs, NK or cross-reactive T cells would compare to vaccinated individuals for protection. Perhaps the authors can discuss on this.

Thank you for this comment. We agree that these measures should be compared with seropositive vaccinated and hybrid immunity cohorts to understand the contribution of these mechanisms on the more complex antigen-experienced background. We are indeed analysing data from a seropositive cohort in a Delta variant breakthrough infection model that will address some of these question. A seropositive challenge study with the same viral strain showed nasal IgA against S and N and predicted epitope (S and non-S) CD8⁺ T cell responses were protective against transient infection (Jackson et al Lancet Microbe, 2024). This suggests that memory T cells and/or antibodies are a primary mechanism protection in this context, although the innate immune data from this study are awaited. We speculate that in the presence of high levels of virus-specific antibody there may also be a shift to a greater reliance on more differentiated cytotoxic and antibody-dependent NK cell functions. We have added this to the discussion lines 549-553.

4. The antibodies tested are against spike or NP, while the cross-reactive T cells are responsive to NSP7, NSP12, and NSP13. Is it possible that pre-existing antibodies against the NSP peptides (or RTC) could play a role in resisting the infection?

Antibodies against the NSPs of ORF1ab have been previously described following confirmed SARS-CoV-2 infection (Shrock et al Science, 2020; Li et al Cell Rep, 2021) but these are rare. As these proteins are not part of the SARS-CoV-2 virion and are only present when transcribed inside an infected cell, it is unlikely that antibodies against the RTC contribute to antiviral immunity. We now state this in the discussion lines 442-446.

5. If no antibodies are involved, and the cDC1/monocytes could be the major antigen presenting cells, would there be sufficient NSP/RTC proteins for presentation when the infection is controlled in a timely manner without the chance for viral replication (or low undetectable level)? Would it mean that these antigens are even more immunodominant than spike proteins or NP?

RTC proteins have been shown to be expressed and presented at detectable levels earlier than structural proteins. Whole-proteome analysis of *in vitro* infected cell lines revealed that early expressed NSP viral proteins contributed more to HLA class I presentation and immunogenicity

than structural proteins. Both NSP12 and NSP13 epitopes were in the top 30 most presented peptides in A549 and HEK293 cell lines after infection with ancestral SARS-CoV-2 infection (2019-nCoV/USA-WA1/2020 isolate) (Weingarten-Gabbay et al Cell, 2021). As RTC proteins are part of the first open reading frame to be expressed in an infected cell, their translation can occur before viral detection by RIG-I that has been shown to act as a restraining factor, pausing viral replication and delaying the generation of subgenomic RNAs and subsequent structural proteins (Yamada et al Nat Imm, 2016). Thus, it is biologically plausible that some cells can present NSPs, be recognised and removed by RTC-specific T cells before structural proteins have been presented.

As suggested by the reviewer, pre-existing memory T cells targeting RTC-specific peptides may therefore control early viral replication before significant viral replication, and RTC-specific T cells have been shown to be uniquely immunodominant only in this type of abortive infection (Swadling et al Nature, 2022), whereas highly expressed structural proteins dominate once sustained infection takes hold and triggers adaptive immunity (Swadling, et al Nature 2022; Tarke et al Cell Rep Med, 2021). There is no evidence to suggest that these antigens are more immunodominant than spike or N but their timing of expression is likely to preferentially stimulate early acting T cells. A sentence has been added to the discussion on this (lines 489-491).

Minor

comments:

1. Line 197: The authors suggest that cDCs could be producing CCL13 once they traffic into the site, could there be CCL13 produced by tissue-resident DCs?

CCL13 was measured in the nasopharyngeal myeloid compartment. However, due to the small numbers of DCs in nasal samples overall, we were unable to determine if these cells were *bona fide* tissue resident or cells that had transited from the circulation. We have now assessed the CD69⁺CD103⁺ DC population, which represents a population with a tissue resident phenotype, using scRNAseq and show the CCL13 and CCL22 expression of this in Figure 3. We speculate a mutual reinforcement between local chemokines and DCs leading to a cycle of trafficking between the blood and tissue, perhaps involving both tissue resident and migratory DCs, and aim to investigate this further in future studies with nasal tissue sampling and timepoints specifically tailored for this question.

2. The role of IL-18, as shown in Extended Data Table 1 should be discussed, as it can enhance NK functions.

Thank you for the suggestion. This has been further addressed in response to a comment from reviewer 1. As mentioned earlier, while the difference in IL-18 levels between infected and uninfected groups did just reach statistical significance without correction, it was indeed higher immediately prior to inoculation in those who developed infection compared with those who resisted infection suggesting a potential detrimental role for IL-18 in the nasal mucosa. Additional text has been added to lines 232-240.

3. Titles for figure legends for Fig. 5 and Extended Data Fig. 3 had formatting issues where it is bulged by the figures.

Thank you, we have made sure this is not the case in the final version.

4. Should Fig. 1b go before 1a?

Thank you for this suggestion. We have made this change.

5. Cluster 1 in Fig. 4a and 4c are described in the text. Did I miss cluster 2?

We apologise for the ambiguity of how some of this section was presented. This has been reanalysed and rephrased in response to the comments from reviewer 1. We now refer only to FlowSOM clusters numbered 1-12 in Figure 4.

6. Extended Data Fig. 3, is a plot missing in panel d first row of FACS plots where the arrow is pointing?

Thank you for pointing this out. The plot was actually present but the arrow should have been pointing to the next plot on the next row; this has been changed.

7. Line 561, the fluorophore for CD3 is missing

Thank you; this has been added.

Reviewer #4 (Remarks to the Author):

Reviewer #5 (Remarks to the Author):

Figures for reviewers only:

[editorial note: figures redacted]

Please find our point-by-point response to the reviewers' comments below.

Reviewer #1 (Remarks to the Author):

All comments have been addressed.
Many thanks, we appreciate your input.

Reviewer #2 (Remarks to the Author):

The authors have addressed all concerns raised in the initial round of reviews and the revised manuscript is appropriate for publication.
Many thanks, we appreciate your input.

Reviewer #3 (Remarks to the Author):

The authors have adequately addressed my comments. The revised manuscript is significantly improved with the additional data and the presentation of them.
Many thanks for this kind feedback, we appreciate your input.

Reviewer #4 (Remarks to the Author):

Reviewer #5 (Remarks to the Author):
